# Managing Ascites and Kidney Dysfunction in Decompensated Advanced Chronic Liver Disease: From "One Size Fits All" to a Multidisciplinary-Tailored Approach

Mario Romeo [1], Carmine Napolitano [1], Paolo Vaia [1], Fiammetta Di Nardo [1,*], Silvio Borrelli [2], Carlo Garofalo [2], Luca De Nicola [2], Alessandro Federico [1] and Marcello Dallio [1]

[1] Hepatogastroenterology Division, Department of Precision Medicine, University of Campania Luigi Vanvitelli, 80138 Naples, Italy; mario.romeo@unicampania.it (M.R.); carmine.napolitano1@studenti.unicampania.it (C.N.); paolo.vaia@studenti.unicampania.it (P.V.); alessandro.federico@unicampania.it (A.F.); marcello.dallio@unicampania.it (M.D.)

[2] Nephrology and Dialysis Unit, Department of Advanced Medical and Surgical Sciences, University of Campania Luigi Vanvitelli, 80138 Naples, Italy; silvio.borrelli@unicampania.it (S.B.); carlo.garofalo@unicampania.it (C.G.); luca.denicola@unicampania.it (L.D.N.)

* Correspondence: fiammetta.dinardo@studenti.unicampania.it; Tel.: +39-081-5664311

## Abstract

Ascites and renal dysfunction are among the most frequent and severe complications of decompensated advanced chronic liver disease (dACLD), often representing two inter-related manifestations of a shared pathophysiological continuum. Recurrent ascites and refractory ascites pose significant therapeutic challenges and are frequently associated with kidney impairment, particularly hepatorenal syndrome. Recent advances have reshaped the understanding of the underlying mechanisms, moving beyond the classical paradigm of peripheral arterial vasodilation to encompass systemic inflammation, gut dysbiosis, and cirrhosis-associated immune dysfunction (CAID). These insights have prompted a shift from uniform treatment protocols toward personalized, multidisciplinary strategies. Therapeutic innovations such as long-term albumin infusion, a transjugular intrahepatic portosystemic shunt, and the Alfapump® system offer promising options, though each requires careful patient selection. Emerging approaches—including fecal microbiota transplantation and peritoneal dialysis—further expand the therapeutic landscape. Ultimately, early risk stratification, the integration of non-invasive tools, and individualized care models are essential to improving outcomes in this high-risk population. This review synthesizes current evidence and highlights future directions for the tailored management of dACLD patients with ascites and renal dysfunction.

**Keywords:** ascites; advanced chronic liver disease; interdisciplinary approach; evidence-based medicine; precision medicine

## 1. Background

### 1.1. Hepatic Recompensation: Revolutionizing the "Irreversibility" in Advanced Chronic Liver Disease

For decades, a biphasic course has been classically proposed to describe the natural history of advanced chronic liver disease (ACLD), recognizing the transition from compensated (cACLD) to decompensated ACLD (dACLD) as a critical and irreversible prognostic turning point [1].

In the compensated stage, liver function is relatively preserved, with patients remaining asymptomatic for various years. However, as portal hypertension and systemic inflammation worsen, hepatic function declines, and the disease progresses to the decompensated phase, burdened by several life-threatening complications [1–3]. In this scenario, ascites represents the most common liver-related event (LRE), marking the earliest clinical manifestation of decompensation in a significant proportion (annual incidence: 5–10%) of cACLD patients every year [4,5]. Importantly, the onset of ascites prognostically constitutes a dramatic watershed, significantly impacting the median survival (dropping from over 10 years in cACLD to less than 2 years in dACLD without effective treatment or liver transplantation), as well as representing the praeludium for further LREs [6,7]. In this sense, patients decompensating with ascites present a higher risk of short-term "non-ascites-related" events (NAREs)—including variceal bleeding and hepatic encephalopathy (HE) [6].

The traditional theory sustaining the irreversibility of the cACLD to dACLD transition has been recently revolutionized, as new evidence proposes the novel concept of "hepatic recompensation," suggesting the possibility of reversing this progression and the relatively dramatic outcomes [8–10]. The control or removal of etiological factors contributing to chronic liver disease (e.g., alcohol abstinence, hepatitis C virus eradication, etc.), simultaneously with biochemically proven preserved liver function (sustained improvement of serum albumin, bilirubin, and INR) and HE resolving in the absence of further episodes of ascites (without active diuretic and/or lactulose/rifaximin treatment) or variceal bleeding for 12 months, has been proposed to define the hepatic recompensation [8–10].

Anyway, despite this encouraging emerging scenario, the recompensation appears to be a possible and desirable outcome exclusively in patients with uncomplicated ascites, continuing to be incompatible in the case of ascites-related complications, including recurrent ascites (RecA) and refractory ascites (RA) [8–10].

### 1.2. Exploring Refractory Ascites: When Hepatic Recompensation Remains Utopian

RecA and RA represent two entities significantly burdening the dACLD clinical course, collectively affecting approximately 10–20% of dACLD patients within 3 years [3,11,12].

RecA is defined as ascites that recurs at least three times within one year despite adherence to dietary sodium restriction and diuretic therapy, representing an early stage or precursor of RA [3]. RA refers to ascites that cannot be mobilized or that recur early (within 4 weeks) after a large-volume paracentesis (LVP) and cannot be prevented by medical therapy [3,11]. In dACLD, RA occurrence has been associated with an elevated 1-year mortality rate, ranging from 30% to 50%, depending on treatment availability, as well as on the RA subtype: diuretic-resistant ascites and diuretic-intractable ascites [3,11].

Diuretic-resistant ascites refers to abdominal fluid accumulation that persists despite the administration of the maximum tolerated doses of diuretics (typically spironolactone up to 400 mg/day and furosemide up to 160 mg/day) combined with dietary sodium restriction (usually ≤2 g/day) for at least one week [3,11]. In this case, the failure to control ascites is due to a lack of therapeutic response and, unlike diuretic-intractable ascites, is independent of the diuretic-related adverse effects [3,11]. Mean weight loss of <0.8 kg over 4 days and daily urinary sodium excretion less than the daily sodium intake defines the lack of response, whereas the reappearance of grade 2 ascites with moderate symmetrical abdominal distension or grade 3 ascites with marked abdominal distension within 4 weeks of initial mobilization refers to "early ascites recurrence" [3,11].

Diuretic-intractable ascites occurs when the use of effective doses of diuretics is precluded by the development of diuretic-induced complications, including, among others, HE, an electrolyte imbalance [hyponatremia (serum sodium < 125 mmol/L), hypokalemia

or hyperkalemia], and renal impairment (e.g., acute kidney injury or rising serum creatinine) [3,11]. Anyway, besides this last-mentioned iatrogenic *primum movens*, kidney dysfunction constitutes a recurrent plague affecting the routine clinical management of dACLD-RA patients [13,14].

### 1.3. Kidney Dysfunction and Ascites in Decompensated Advanced Chronic Liver Disease

Kidney dysfunction represents a complex clinical and therapeutic challenge frequently observed in dACLD RA-affected patients, and, particularly, RA has been *per se* associated with a heightened risk of developing hepatorenal syndrome (HRS) [6,13,14]. Notably, the occurrence of HRS has been reported in a significant proportion (~20%) of dACLD-RA affected patients, ultimately contributing to increasing the risk of hospitalization and worsening outcomes (estimated mortality rate ~50%) in this population [11,12,15]. Altogether, these data highlight the urgent need for early recognition.

Anyway, although uncomplicated ascites, RecA and RA, and kidney dysfunction are defined as distinct clinical entities, in practice, they often represent a continuum of disease severity [11,12,15]. Over 5 years, approximately 10% of patients with uncomplicated ascites progress to RA, and among these individuals, up to 40% may develop kidney dysfunction within the same timeframe [11,12,15].

Simultaneously, physiopathological and routine clinical practice reasons fuel this continuum. On one side, indeed, dACLD patients presenting RA require multiple and repeated LVPs with an intrinsic higher risk of hospitalization and nosocomial infections, potentially precipitating organ dysfunction, including renal performance [16]. On the other hand, portal hypertension and systemic inflammation synergistically determine splanchnic vasodilation with a reduced effective arterial volume, representing physiopathological common denominators sustaining liver disease progression, ascites (re)occurrence, and kidney dysfunction [17].

However, in the intertwined and reciprocal relationship between ascites worsening and renal performance impairment in dACLD, the proper identification of distinct kidney dysfunction phenotypes is crucial, considering the different prognostic repercussions and the required management strategies [18].

A decade ago, the concept of kidney dysfunction in dACLD shifted away from relying on a static serum creatinine cutoff (e.g., >1.5 mg/dL), emphasizing dynamic changes in creatinine levels and recognizing that even small increases can be clinically significant in cirrhotic patients [3,19]. This approach has enabled a more nuanced understanding of renal dysfunction in liver disease, focusing on trends rather than absolute thresholds [3,19].

Historically, HRS was initially categorized into two distinct types: HRS type 1 and HRS type 2, based on the rapidity of renal function deterioration and prognosis [19,20].

HRS type 1 was defined by a rapid and progressive decline in renal function, often leading to acute kidney injury (AKI) with a very poor prognosis, while HRS type 2 involves a more gradual and moderate reduction in the glomerular filtration rate, typically associated with RA [19,20]. This dichotomy, while foundational, occasionally presented diagnostic ambiguities, particularly in differentiating HRS type 1 from other forms of AKI in cirrhotic patients [19,20].

For this purpose, a subsequent transition to a more modern viewpoint, distinguishing between HRS-non-acute kidney injury (NAKI) and HRS-AKI, aimed to refine diagnostic precision and therapeutic stratification by emphasizing the acute nature of renal impairment [3,21–23]. This conception confirmed that HRS-NAKI is a more chronic form, developing gradually over several weeks. This entity is associated with the progressive deterioration of systemic hemodynamics, leading to a sustained increase in renal vasoconstriction and a slower decline in kidney function [3,21–23]. In particular, HRS-NAKI

was furthermore divided into different subtypes according to the Kidney Disease Global Outcome (KDIGO) criteria: HRS acute kidney disease (HRS-AKD) if the estimated glomerular filtration rate (eGFR) is reduced (<60 mL/min/1.73 m$^2$) for less than three months and HRS-chronic kidney disease (HRS-CKD) if the reduction of eGFR persists for a longer period (i.e., >3 months) [3,21–23]. Conversely, HRS-AKI represents the acute form of the syndrome, characterized by a rapid decline in kidney function over a few days. This entity typically occurs in response to an acute event that further precipitates the abnormal renal hemodynamics, featuring the advanced liver disease scenario [3,21–25].

In line with this, in dACLD patients, HRS-AKI develops exclusively in patients presenting with ascites, as recently restated by the Acute Disease Quality Initiative (ADQI) and International Club of Ascites (ICA) joint multidisciplinary consensus meeting. In particular, according to this novel statement, the occurrence of HRS-AKI should be established based on the following criteria: (a) presence of ACLD complicated by ascites; (b) a rise in serum creatinine exceeding 0.3 mg/dL within 48 h, or an increase of more than 50% from the known or presumed baseline occurring within the previous 7 days; alternatively, urinary output less than 0.5 mL/kg/h sustained for over 6 h; (c) lack of renal function improvement (serum creatinine and/or urinary output) within 24 h following adequate intravascular volume expansion; (d) no compelling evidence of an alternative etiology responsible for the AKI episode [26].

Finally, going beyond the "HRS-AKI" and "HRS-NAKI" dualism, the ADQI and ICA joint multidisciplinary consensus also proposed a contemporary and more dynamic definition of kidney dysfunction phenotypes, proposing AKI, AKD, and CKD as a pathophysiological continuum in which an initial renal insult may culminate in either functional recovery via adaptive repair, sustained renal impairment, or progression to CKD through maladaptive reparative mechanisms [26]. AKI is encompassed within the broader AKD spectrum; consequently, all individuals diagnosed with AKI inherently meet the criteria for AKD. Recurrent AKI episodes may manifest throughout the clinical trajectory of a single patient. Even following apparent AKI resolution, residual structural and/or functional renal abnormalities may persist, thereby fulfilling the AKD diagnostic criteria [26]. Patients with HRS–AKD who concurrently satisfy AKI criteria are reclassified as HRS-AKI [26].

The last-mentioned updated definition was proposed to better reflect the pathophysiological continuum and clinical urgency associated with the rapid decline in kidney function, moving away from a solely prognostic classification to one that guides immediate management strategies [17,25]. Anyway, despite this proposed novel viewpoint, the management of AKI is still largely based on evolving guidelines where the "HRS-AKI" and "HRS-NAKI" dualism continues to exist, and a unified, standardized approach remains unestablished [27].

Figure 1 reports the close relationship between ascites worsening and kidney dysfunction in the dACLD scenario, summarizing the main features of the different ascites and renal impairment clinical phenotypes (Figure 1).

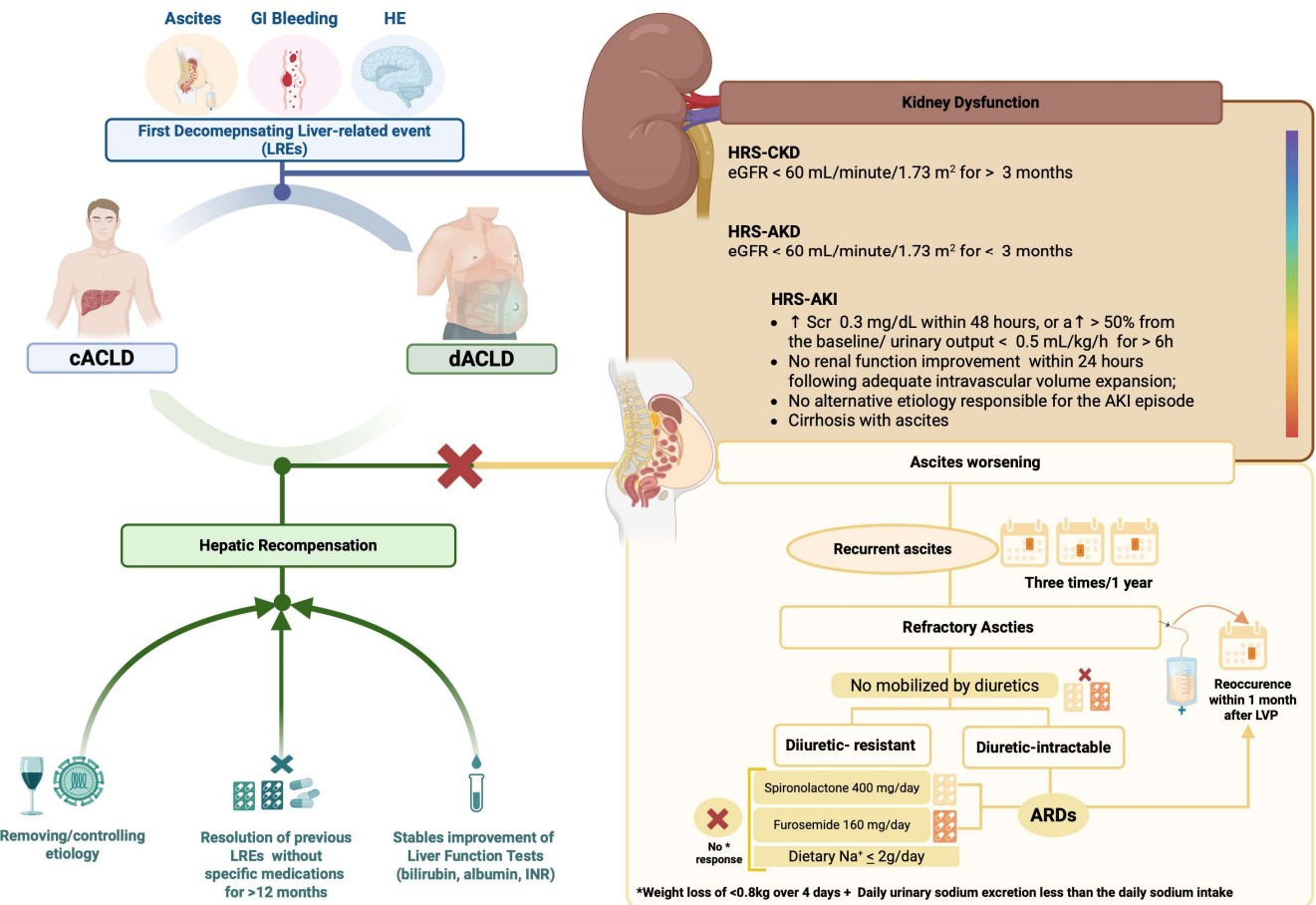

**Figure 1.** Ascites and kidney dysfunction in decompensated advanced chronic liver disease: subtypes and phenotypes. cACLD: compensated advanced chronic liver disease; dACLD: decompensated advanced chronic liver disease; LREs: liver-related events; HRS: hepatorenal syndrome; LVP: large volume paracentesis; ARDs: adverse-reaction drugs; AKI: acute kidney injury; CKD: chronic kidney disease; AKD: acute kidney disease; Scr: serum creatinine; eGFR: estimated glomerular filtration rate; INR: international normalized ratio; h: hour(s); ↑: increased. Figure 1 was created by using the web graph software BioRender® https://app.biorender.com/user/signin.

### 1.4. Kidney Dysfunction and Ascites in Acute-on-Chronic Liver Failure (ACLF)

Acute-on-chronic liver failure (ACLF) represents a well-recognized syndrome potentially burdening the clinical course of dACLD patients presenting acute decompensation [28], typically characterized by the rapid deterioration of liver function, systemic inflammation, and the failure of one or more extrahepatic organs [29]. Notably, in this scenario, ACLF is associated with high short-term mortality and represents a clinical state distinct from "mere" decompensation [29].

According to the European Association for the Study of the Liver (EASL)-Chronic Liver Failure (CLIF) Consortium, ACLF can be diagnosed using the CLIF-Sequential Organ Failure Assessment (CLIF-SOFA) score and categorized into three grades based on the number of organ failures: grade 1 (a single organ failure, usually renal), grade 2 (two organ failures), and grade 3 (three or more organ failures) [28–30]. Relevantly, mortality increases substantially with each additional organ failure, making the early identification and classification essential for prognosis and management [28].

Kidney dysfunction represents a common and pivotal dramatic moment contributing to the configuration of ACLF onset and worsening.

In this sense, among organ failures featuring ACLF, renal failure is the most frequent and has the greatest impact on prognosis [17].

In addition, the presence of kidney failure alone is sufficient to define ACLF grade 1, especially when associated with moderate-to-severe HE or non-renal organ dysfunctions [28–30]. Furthermore, in most cases, renal failure in ACLF presents as HRS-AKI, with severe short-term prognostic repercussions [17]. Finally, renal impairment is in turn able to precipitate and worsen failures in other organs, including the liver and coagulation systems, thus accelerating the downward clinical trajectory [17]. In light of this, kidney function emerges as a central therapeutic target in ACLF management, and its deterioration often signals the need for advanced interventions or intensive care support in these patients, particularly when ascites coexists [31].

In particular, the development of RA in patients with ACLF often coexists with renal dysfunction, suggesting significant impairment in effective arterial blood volume and the worsening of circulatory failure [31]. Moreover, the occurrence of infection in ascitic fluid can configure spontaneous bacterial peritonitis (SBP), which, in turn, represents a major precipitating factor for ACLF and can rapidly lead to multiorgan failure [32].

Concerning this, cornerstone research on this topic revealed the presence of ascites as a clinical variable independently associated with an increased risk of renal failure following bacterial infections, which in turn represents a key precipitating event for ACLF [28,33,34].

Therefore, although ascites itself is not included as a defining criterion in the CLIF classification of ACLF, this manifestation of decompensation represents a crucial clinical indicator of disease progression [28,31].

Anyway, ascites in ACLF should not be intended as a mere marker of severity, representing, in addition, a crucial contributor to worsening hemodynamic instability and further organ dysfunction [29,31]. Consequently, the presence of ascites at baseline should prompt heightened clinical surveillance for ACLF progression, particularly in the setting of infections [3,31].

### 1.5. Aim of the Present Research

Despite growing research efforts to classify and characterize the phenotypes of kidney dysfunction in patients with dACLD and ascites, a significant portion of this large topic remains unexplored.

The dramatic mortality burdening the clinical course of patients with RA and kidney dysfunction is driven by an incompletely clarified physiopathological mechanism, of which the complete elucidation continues to represent a challenge. Moreover, the evolving understanding of HRS and its subtypes, as well as the prognostic relevance of the early identification of patients developing AKI, highlights the complexity of this clinical landscape, suggesting the need for tailored multidisciplinary approaches.

Considering this background, after rapidly reviewing the main common physiopathological moments simultaneously sustaining RA and renal dysfunction in dACLD, the present research provides a status of the art of the current therapeutic strategies, remarking on the absolute need for implementing future personalized approaches in the management of these complex and interrelated diseases.

## 2. Overweening Physiopathology of Ascites and Kidney Dysfunction

### 2.1. From "Classic" Peripheral Arterial Dilatation to the Novel "Cirrhosis-Associated Immune Dysfunction (CAID)" Theory

Historically, the development of ascites and ascites-related events (including HRS, dilutional hyponatremia, and SBP) in ACLD has been attributed to the severe sinusoidal portal hypertension and hepatic insufficiency (including decreased albumin production), collectively initiating a cascade of hemodynamic and neurohormonal changes [3,35,36].

In particular, the peripheral arterial vasodilation hypothesis posited that splanchnic vasodilation, driven by portal-hypertension-induced increased release of nitric oxide and other vasodilators, led to effective arterial hypovolemia, triggering neurohumoral activation—including the renin–angiotensin–aldosterone system (RAAS), sympathetic nervous system (SNS), and antidiuretic hormone (ADH) release [3,35–38]. The activation of these systems classically promotes sodium and water retention, contributing to the accumulation of fluid in the peritoneal cavity (ascites formation) and ascites-related events (AREs) (including dilutional hyponatremia and kidney dysfunction) [3,35–38]. In advanced stages, along with the perpetuation of these mechanisms, fluid retention becomes resistant to diuretics, promoting, on one side, RecA/RA onset, as well as, on the other, a marked reduction in the renal perfusion, thus predisposing these patients to HRS [39].

Therefore, in this apparently "simple" scenario, worsening ascites represents more than the mere epiphenomenon of fluid overload and has to be intended as a manifestation of complex systemic and renal hemodynamic disturbances, where the interplay among portal hypertension, neurohormonal activation, and renal hypoperfusion is crucial [38–40].

Besides this classic paradigm, in the last decade, growing scientific efforts have focused on the identification of other pathogenetic actors sustaining these mechanisms and the relative perpetuation driving ascites worsening and kidney dysfunction, progressively evolving from the classical paradigm of peripheral arterial vasodilation to a more nuanced understanding centered on systemic inflammation [41,42].

On one side, in a cornerstone study on this topic, Dirchwolf et al. initially reported increased serum levels of proinflammatory cytokines [Interleukin (IL-) 6, IL-8, and Tumor necrosis factor-alpha] in decompensated cirrhotic patients compared with stable cirrhotic and healthy controls in the absence of any evidence of bacterial infection, also highlighting a positive correlation between IL-6/IL-8 and hepato-functional status defined by the Model for End-Stage Liver Disease (MELD) score [43].

On the other hand, further research focused on the identification of the source potentially sustaining systemic inflammation in this setting. About this, various evidence revealed an increase in intestinal permeability simultaneously with altered gut microbiota composition and function in patients with ACLD, as well as the association of an altered profile of the intestinal microbiome with systemic endotoxemia [44–46], ultimately supporting the role of "gut-microbiota induced systemic inflammation" as the novel and complementary driver of decompensation together with portal hypertension worsening [41,42].

According to the modern theory, portal hypertension and gut dysbiosis reciprocally contribute to impairing intestinal permeability, promoting the translocation of bacteria and microbe-derived products, including lipopolysaccharide (LPS) (endotoxemia) [41,42]. Relevantly, bacterial translocation (BT) [47], even in the early stages of disease, initiates a cascade of immune activation via pathogen-associated molecular patterns (PAMPs), notably LPS and bacterial DNA, which correlate with elevated levels of pro-inflammatory cytokines such as TNF-$\alpha$ and IL-6 [3,41,42]. The perpetuation of this inflammatory milieu exacerbates endothelial dysfunction (worsening splanchnic vasodilatation and renal hypoperfusion), impairs renal performance, and contributes to the development of complications (including RA and HRS-AKI) [48].

Interestingly, Bernardi et al. initially revealed a parallel trend between the increasing severity of BT (moderate BT, significant BT, and severe BT), liver disease progression stages (cACLD, early dACLD, and end-stage dACLD), and the worsening of portal hypertension, with the severity of systemic inflammation, effective hypovolemia, and the risk of renal failure occurrence [41]. In support of this model, subsequent research has confirmed the progressive increase in systemic inflammation markers, including C-reactive protein (CRP) and IL-6, along the cACLD–dACLD progression, also highlighting their independent asso-

ciation with decompensation and mortality [49]. Furthermore, non-selective beta-blocker (NSBB) administration, by impacting portal hypertension severity, has been demonstrated to determine systemic anti-inflammatory effects, reducing CRP levels, correlating with improved clinical outcomes [50].

Altogether, these findings suggest that systemic inflammation is not merely a consequence and has to be intended as a main driver of disease progression, simultaneously with portal hypertension worsening in dACLD, warranting its integration into future prognostic tools, therapeutic strategies, and extended pathogenetic models [41,42]. In this sense, adrenergic system dysfunction represents another relevant factor playing, in concert with systemic inflammation, a relevant role in promoting AREs [3,41,42,50].

In dACLD, suppression of the hypothalamic–pituitary–adrenal (HPA) axis, reduced blood volume, impaired cholesterol synthesis, and the release of inflammatory cytokines have been shown to simultaneously interfere with adrenal steroidogenesis [3,41,42,50]. This hormonal imbalance leads to heightened sympathetic nervous system activity, which in turn causes several downstream effects: (1) proximal tubular sodium retention and diuretic resistance, making fluid management more difficult [51]; (2) worsening of cardiovascular dysfunction [52,53], due to impaired circulatory regulation, contributing to the reduction of effective volume, and thus kidney dysfunction; (3) altered intestinal motility promoting increased BT, further fueling systemic inflammation by abnormally activating innate immune cells, ultimately configuring a complex pathogenetic vicious circle [3,41,42,50]. Focusing on cardiovascular dysfunction in dACLD, a specific cardio-functional impairment has been proposed in the ACLD setting, where, importantly, a critical contributor to renal injury appears to be represented by "high-output" cardiac failure [52,53]. This condition is defined by a hyperdynamic circulatory state characterized by elevated cardiac output, reduced systemic vascular resistance, and impaired tissue perfusion [52,53]. Although cardiac output is increased, the effective arterial blood volume remains insufficient due to the above-mentioned portal-hypertension-related and systemic-inflammation-determined splanchnic vasodilation, ultimately resulting in a paradoxical scenario where renal perfusion is compromised despite an ostensibly robust circulatory flow [52,53]. This pathophysiological mismatch places chronic strain on the cardiac fibers and, over time, the perpetuation of this phenomenon can lead to structural remodeling and functional alterations, collectively termed "cirrhotic cardiomyopathy" ("CCM") [3,52,53]. CCM includes blunted contractile responsiveness to stress, diastolic dysfunction, and electrophysiological abnormalities, collectively contributing to further cardio-functional impairment, progressively reducing the cardiac ability to adapt to systemic circulatory demands. Therefore, the recognition of high-output cardiac failure as a distinct pathophysiological entity in ACLD underscores the need for individualized hemodynamic assessments, considering also the relative therapeutic repercussions: in this sense, this peculiar cardiovascular profile also complicates treatment strategies, since aggressive volume expansion may worsen cardiac strain, while vasopressor use must be carefully titrated to avoid exacerbating renal ischemia or precipitating cardiac decompensation [3].

Relevantly, emerging evidence has suggested the role of BT-determined chronic low-grade endotoxemia as a factor directly affecting heart functioning, highlighting the LPS implications in inducing myocardial inflammation (via TLR4 signaling), promoting contractile dysfunction and impaired vascular tone regulation, ultimately exacerbating the typical ACLD hyperdynamic circulatory state [54,55].

Collectively, the progressive clarification of these interlinked mechanisms led to the conception that dACLD complicated by AREs is more than a mere hemodynamic disorder and should be intended as a complex systemic syndrome involving immune dysregu-

lation [56], opening the doors to the last pathogenetic frontier of "Cirrhosis-Associated Immune Dysfunction" (CAID) [57–59].

Cirrhosis-Associated Immune Dysfunction (CAID) represents a dynamic and multi-faceted immunological syndrome intrinsic to dACLD progression [57,58].

CAID encompasses a spectrum of immune alterations characterized by the coexistence of systemic inflammation and immune deficiency, both of which intensify with disease severity and are pivotal in determining clinical outcomes [58]. The pathophysiological basis of CAID involves dysregulated innate and adaptive immune responses, largely driven by BT and chronic endotoxemia, which, by acting as PAMPs, mediate the persistent activation of pattern recognition receptors (PRRs) such as Toll-like receptors (TLRs) [57,58,60]. In this scenario, while initially, circulating immune cells show the increased expression of activation markers and pro-inflammatory cytokines (including TNF-α, IL-6, and IL-1β), monocytes, macrophages, and neutrophils subsequently exhibit impaired phagocytic and bactericidal functions as a consequence of persisting activation ("immune exhaustion") [61]. According to this, two distinct inflammatory phenotypes have been described: a low-grade systemic inflammatory state featuring the initial phase, and a high-grade inflammatory phenotype associated with ACLF [28,62], marked by immune paralysis, organ failure, and high short-term mortality, characterizing the advanced stages of the disease [57,61].

Relevantly, the immunocompromisation predisposes patients to life-threatening infections (e.g., SBP) [63], which are major contributors to multiorgan failure (including renal failure), determining a dramatic increase in hospitalization and mortality [62].

Therefore, considering the relevance of this emerging theory, the further elucidation of pathogenetic mechanisms featuring the immunological landscape of CAID represents a crucial research challenge for developing tailored prognostic biomarkers and future personalized therapeutic interventions in dACLD patients based on specific individual immune profiles [64].

Figure 2, by summarizing the most relevant hemodynamic and systemic inflammation-related features (Figure 2A), reports the main physiopathological mechanisms sustaining ascites worsening and kidney dysfunction in dACLD, including the major highlights characterizing the CAID phenotypes (Figure 2B). Figure 3 summarizes and integrates the physiopathological crucial moments of the "classic" vasodilatation theory with the emerging CAID-related mechanisms (Figure 3).

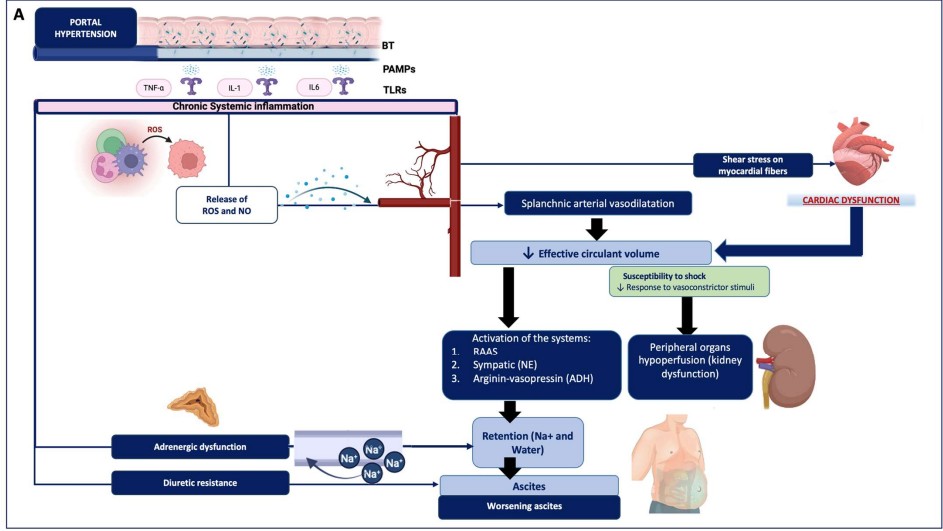

**Figure 2.** *Cont.*

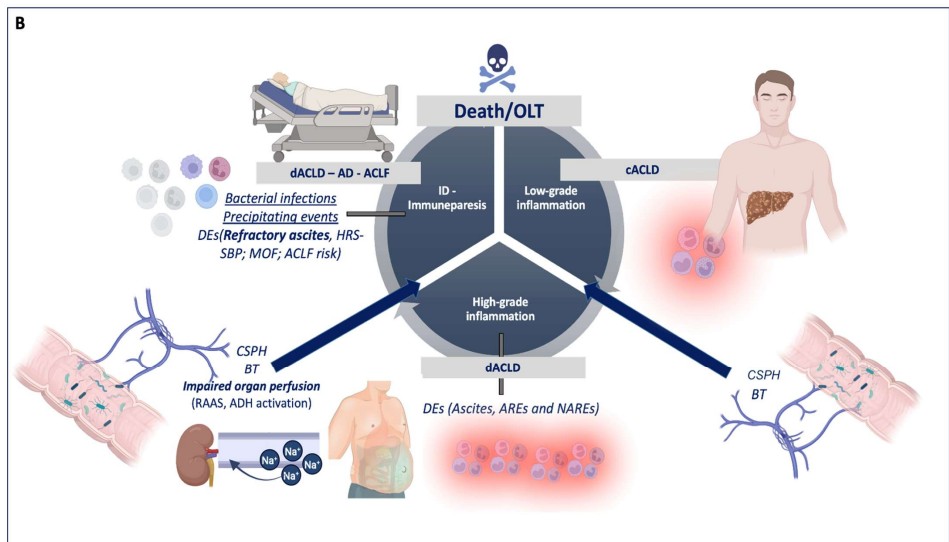

**Figure 2.** Physiopathological drivers of ascites and kidney dysfunction in decompensated advanced chronic liver disease (**A**): main hemodynamic alterations and systemic-inflammation-related mechanisms contributing to complicating ascites and kidney dysfunction in dACLD. (**B**): major highlights of Cirrhosis-Associated Immune Dysfunction (CAID) and relative phenotypes. BT: bacterial translocation; TLRs: toll-like receptors; PAMPs: pathogen-associated molecular patterns; IL: interleukin; TNF: tumor necrosis factor; ROS: reactive oxygen species; NO: nitric oxide; NE: norepinephrine; ADH: antidiuretic hormone; renin–angiotensin-aldosterone system; Na+: sodium; OLT: orthotopic liver transplant; cACLD: compensated advanced chronic liver disease; dACLD: decompensated advanced chronic liver disease; AREs: ascites-related events; NAREs: non-ascites related events; CSPH: clinically significant portal hypertension; HRS: hepatorenal syndrome; DEs: decompensating events; AD: acute decompensation; MOF: multiorgan failure; SBP: spontaneous bacterial peritonitis; ACLF: acute-on-chronic liver failure; ↓: decreased. Figure 2 was created by using the web graph software BioRender® https://app.biorender.com/user/signin.

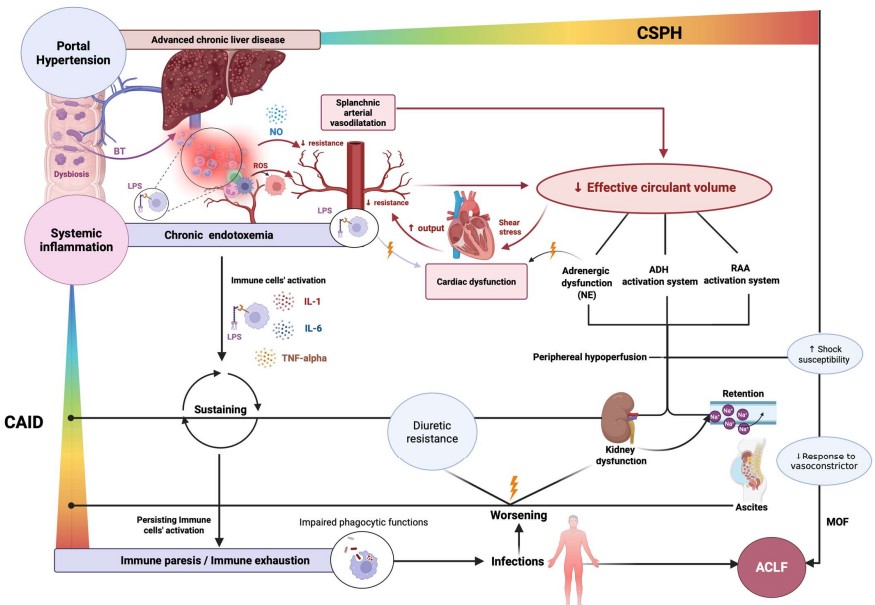

**Figure 3.** The "classic" (hemodynamic-related) and "modern" (immune-related) physiopathological drivers of ascites and kidney dysfunction. Cirrhosis-Associated Immune Dysfunction (CAID). BT: bacterial translocation; IL: interleukin; TNF: tumor necrosis factor; ROS: reactive oxygen species; NO: nitric oxide; LPS: lipopolysaccharide; NE: norepinephrine; ADH: antidiuretic hormone; renin–angiotensin–aldosterone system; Na+: sodium; CSPH: clinically significant portal hypertension; MOF: multiorgan failure; ACLF: acute-on-chronic liver failure; ↑: increased; ↓: decreased. Figure 3 was created by using the web graph software BioRender® https://app.biorender.com/user/signin.

*2.2. Systemic Inflammation: One Shared Pathogenetic Driver for Two Divergent Clinical Trajectories: Elucidating the Paradox of "Ascites Worsening and Variceal Regression"*

The reconceptualization of chronic inflammation and gut–liver axis dysfunction, simultaneously with portal hypertension worsening, as central and common drivers of liver disease progression, sustaining both AREs and NAREs (including variceal bleeding), represents consolidated evidence [41,42]. However, recurrently, in a certain and non-negligible proportion of dACLD patients, the worsening of ascites is observed in contrast to a regression in the severity of esophageal varices (and relatively risk of bleeding) [6,7], configuring a divergent paradoxical scenario of which the pathophysiological explanation remains complex and heterogeneous [65]. This divergence has been proposed to mainly reflect compartmentalized hemodynamic responses to systemic inflammation ("compartment-specific" inflammation) and differential vascular remodeling [65,66], in a multifaceted pathogenetic context distinguishing the peritoneal compartment, renal circulation, and portal flow.

Relevantly, in the peritoneal compartment, chronic inflammation, by promoting endothelial dysfunction, capillary leakage, and lymphatic impairment, favors fluid accumulation independent of the absolute portal pressure. In line with this, emerging evidence also suggests the compartmentalization of BT and altered immune responses in dACLD. About this, in their research, Alvarez-Silva et al. reported significantly higher levels of IL-6 in ascites fluid compared to blood samples in all dACLD patients, as well as a significantly higher bacterial richness in ascites compared to the corresponding patient's blood [66]. In addition, in the renal circulation, inflammation-driven neurohormonal activation—particularly involving the RAAS—exacerbates sodium retention and renal hypoperfusion, further fueling ascites formation [35,41,67]. In contrast, variceal development (and worsening) appears to be more tightly linked to structural changes in portal venous flow (and intrahepatic resistance), which may plateau or regress under optimized medical therapy [35,41,67]. In this sense, indeed, simultaneously with the above-mentioned physiopathological features, advances in early screening, the widespread use of NSBBs (particularly carvedilol), and proactive endoscopic surveillance represent co-existing factors routinely contributing to the stabilization or regression of varices despite ongoing portal hypertension [50,65].

Therefore, from a translational point of view, further research addressing a deeper investigation of all these (and other unexplored) mechanisms sustaining the dualistic disease course in certain dACLD individuals would simultaneously offer relevant management implications. The identification of novel potential biomarkers specifically differentiating "inflammatory" and "hemodynamic" drivers could guide personalized treatment, as well as allow clinicians to precisely predict the outcomes. In this sense, future clinical trials should stratify endpoints by phenotype (ascitic vs. variceal) to better capture the therapeutic efficacy of the relatively unexplored strategies.

## 3. Therapeutic Options in Managing Ascites and Kidney Dysfunction in Decompensated Advanced Chronic Liver Disease

*3.1. Role of Human Albumin Administration: State of the Art*

3.1.1. The Pleiotropic Effects of Human Albumin: More than a Plasma Volume Expander

In the last decade, the administration of human albumin (HA) in patients with dACLD complicated by ascites and renal dysfunction has emerged as a multifaceted therapeutic strategy that extends beyond its traditional role as a plasma volume expander [68]. Albumin, the predominant plasma protein synthesized by hepatocytes, exerts potent oncotic pressure, thereby stabilizing the intravascular volume and mitigating effective hypovolemia—a hallmark of ACLD [69]. However, its non-oncotic properties, including antioxidant, immunomodulatory, and endothelial-stabilizing functions, are increasingly

recognized as pivotal in counteracting the systemic inflammation and circulatory derangements characteristic of dACLD [70]. Mechanistically, albumin binds and neutralizes PAMPs, thereby attenuating Toll-like receptor-mediated immune activation and oxidative stress [70]. This immunomodulatory effect is particularly relevant in ACLD, where BT and systemic inflammation contribute to progressive organ dysfunction [70].

Relevant clinical trials, including the ANSWER study, have demonstrated that long-term albumin infusion (ANSWER trial dosage: HA 40 g weekly) significantly improves transplant-free survival and reduces the incidence of AREs and NAREs (including SBP, HRS, and HE), ultimately decreasing hospitalizations [71–73]. Also, in the context of SBP, albumin co-administered with antibiotics has been shown to reduce the incidence of AKI and mortality compared to antibiotics alone, as demonstrated in a landmark study [74].

The protective renal effects of albumin are attributed not only to volume expansion but also to its ability to restore renal blood flow autoregulation and improve endothelial function [70]. This is evidenced by reductions in von Willebrand factor and serum nitrite levels, alongside improved creatinine clearance and sodium excretion. Furthermore, albumin may enhance the efficacy of diuretics in patients with RAs by improving the effective arterial blood volume and reducing neurohormonal activation [75].

Despite these promising outcomes, subsequent trials such as ATTIRE and MACHT have raised questions about the generalizability of albumin's benefits, particularly in hospitalized patients with acute decompensation [76–78]. These studies evidenced that albumin's efficacy may depend on the patient selection, timing, and dosing strategy. Additionally, structural modifications of albumin in ACLD patients—such as oxidation and glycation—may impair its functional capacity, highlighting the need for biomarkers to guide personalized therapy [79].

Taken together, the pleiotropic effects of albumin underscore its therapeutic potential as a multitarget agent in ACLD, particularly in patients with ascites and renal impairment. On this topic, more recently, an international position statement addressed the heterogeneous and often conflicting evidence surrounding the use of HA in the management of dACLD-related complications through a structured three-round Delphi process [80]. Short-term HA infusion was confirmed for HRS, LVP, and SBP, while acknowledging the need for further investigation into its role in other dACLD-related complications; on the other hand, long-term HA administration was proposed to be considered in selected settings, where budget and logistical issues can be resolved [80]. Importantly, pulmonary edema emerged as a key safety concern, warranting close monitoring [80].

Collectively, these recommendations offered a refined framework for the clinical use of HA, though optimal timing and dosing strategies remain to be clarified.

Future research should aim to identify responders through molecular profiling and explore synergistic combinations with other disease-modifying agents to optimize outcomes in this vulnerable population [79].

3.1.2. Re-Evaluating the Best Timeframe for Human Albumin Administration in HRS-AKI

HA administration traditionally represents a cornerstone in the management of dACLD-RA/RecA patients developing AKI [3]. In line with this, for many years, the EASL-AKI algorithm proposed HA administration (at a dosage of 1 g/kg/day) for 48 h as a requisite for the diagnosis of HRS-AKI [3]. More recently, the ADQI-ICA joint multidisciplinary consensus on the management of AKI in ACLD patients, by revising the diagnostic criteria, has revolutionized the time of HA administration, recommending against the systematic administration of HA for 48 h as a requisite for the diagnosis of HRS-AKI, restricting the use of albumin infusion from 48 to 24 h [26].

In this sense, albumin infusion (20–25% HA infusion) in combination with vasoconstrictor therapy (terlipressin as first-line agent), should be initiated immediately upon establishing a diagnosis of HRS-AKI according to the novel previously reported criteria, which include the absence of an improvement in serum creatinine and/or urine output within 24 h following adequate volume resuscitation [26].

Rather than the risk of volume overload in a setting where cardiac dysfunction remains very common [52,81], the real major concern guiding the proposal to reduce the albumin infusion to 24 h was the potential delay in starting other effective treatments (e.g., terlipressin) [26].

Anyway, while based on valid scientific rationale, emerging evidence has rechallenged these initial positions, proposing new and more solid data.

Interestingly, Schleicher et al. demonstrated that a considerable proportion (~25%) of patients with dACLD and AKI responded to albumin therapy only in the second 24 h period of treatment in a large multicenter cohort population study.

Moreover, the authors evaluated three different response definitions after 24 and 48 h [(1) serum creatinine decrease >0.3 mg/dL, (2) serum creatinine decrease >25%, and (3) serum creatinine decrease in at least one AKI stage, which was consistent with the proposed EASL-AKI algorithm], and prolonged the follow-up until liver transplantation (LT), death, or hemodialysis.

Notably, in assessing the prognostic significance of these three different definitions of response, the EASL-AKI algorithm-based one was associated with higher hemodialysis- and transplantation-free survival rates [82].

These results were consistent with previous observations revealing how shortening the albumin challenge from 48 to 24 h may lead to the overdiagnosis of HRS-AKI and overtreatment with terlipressin [83]. In detail, in the study by Ma et al., the percentage of patients who required 48 h to respond to albumin infusion was considerable (~40%) [83]. At the same time, the authors, aiming to validate the EASL-AKI algorithm in clinical practice, revealed that a significant proportion (74%) of the non-responders were patients with AKI phenotypes other than HRS-AKI, with a major predominance of acute tubular necrosis (ATN), and that the rate of response to terlipressin in patients meeting HRS-AKI criteria was 61% [83].

These results highlighted the importance of differentiating HRS-AKI from ATN-AKI using the classic criteria and addressed the initial concern regarding the delay in starting other treatments. Indeed, among non-responders to albumin, the median time from AKI diagnosis to terlipressin initiation was only 2.5 days [83].

Collectively, consistently with a recent brilliant expert opinion by Angeli et al. on this topic [84], all these new findings suggest that following the EASL-AKI algorithm remains the best choice to manage HRS-AKI and that shortening the duration of albumin therapy may lead to overdiagnosis and overtreatment with terlipressin.

### 3.2. Role of Terlipressin Administration in dACLD and Ascites: Beyond Hepatorenal Syndrome

Terlipressin represents a well-defined synthetic vasopressin analogue with a relevant role in improving renal function among dACLD patients with ascites by acting on different physiopathological targets [85]. Mechanistically, terlipressin acts primarily on vasopressin receptors (V1 receptors), inducing splanchnic vasoconstriction, thus reducing portal hypertension and contributing to blood flow redistribution toward "noble" organs, including the kidneys [86]. As a consequence, this effect enhances renal perfusion and mitigates the vasodilatory state characteristic of ACLD. Additionally, terlipressin has been shown to suppress RAA overactivation, thereby improving the sodium balance and promoting natriuresis [85,86].

Considering this evidence, HA plus terlipressin administration via i.v. bolus (at the initial dose of 1 mg every 4–6 h) has been proposed by EASL Clinical Practice Guidelines (CPGs) as the first-line therapeutic option for the treatment of HRS-AKI [3], recommending, in the case of non-response (decrease in serum creatinine < 25% from the peak value), after two days, an increase in the dosage in a stepwise manner to a maximum of 12 mg/day [3]. In contrast with bolus i.v. administration, administering terlipressin via continuous i.v. infusion (at an initial dose of 2 mg/day) represents a smart strategy to reduce the global daily dose and, thus, the rate of adverse effects (including, among others, dyspnea, cardiac ischemia, and QT prolongation) [3]. In support of this, clinical trials have demonstrated that continuous terlipressin infusion significantly reduces plasma renin and aldosterone levels while increasing Angiotensin-converting enzyme 2 (ACE2) activity, suggesting an additional favorable modulation of neurohormonal pathways [87].

Relevantly, subsequent research has confirmed its efficacy in improving renal biomarkers even in patients without overt HRS, supporting the expanding role of terlipressin not only in acute settings [19,87]. In this sense, while terlipressin has long been established as a cornerstone therapy for HRS, emerging evidence suggests that its utility may extend to ACLD patients with ascites in the absence of HRS, collectively underscoring the relatively multifaceted role of continuous terlipressin infusion in improving renal function, stabilizing hemodynamics, and mitigating complications associated with RA [88].

On this topic, in a systematic review encompassing 12 studies including randomized controlled trials and observational cohorts, Bai et al. assessed the efficacy of terlipressin in non-HRS dACLD patients with ascites [89]. Notably, terlipressin was shown to prevent paracentesis-induced circulatory dysfunction (PICD) in multiple trials, likely through its vasoconstrictive action on splanchnic vessels and suppression of plasma renin activity. These findings suggest that terlipressin may serve as a protective agent during LVP, a common intervention received by dACLD patients. Interestingly, the authors also reported consistent improvements in renal parameters, including the glomerular filtration rate (GFR), urinary sodium excretion, and serum creatinine levels [89].

Complementing this, Shen et al. provided mechanistic insights and a forward-looking clinical appraisal of the role of terlipressin in ascites management [90]. On one hand, the authors emphasized the dual action of this drug: splanchnic vasoconstriction via V1 receptor activation and renal tubular modulation through V2 receptor engagement. This dual mechanism simultaneously improves renal perfusion and enhances diuresis and sodium retention [90]. On the other side, this research further highlighted recent clinical trials demonstrating reductions in ascites severity, the decreased need for repeated paracentesis, and the stabilization of renal function in patients with borderline renal impairment [90]. Importantly, the authors advocated for individualized dosing strategies and proposed future research directions (including comparisons with midodrine and satavaptan) and the exploration of ambulatory infusion protocols for transplant candidates [90].

Collectively, these studies support a paradigm shift in the therapeutic use of terlipressin. Rather than being confined to HRS, terlipressin may offer broader benefits in the management of ascites and renal dysfunction in dACLD.

In this sense, its ability to modulate systemic and renal hemodynamics, prevent PICD, and improve fluid balance positions it as a promising adjunctive therapy in patients with refractory ascites or those at risk of renal decompensation. However, further high-quality randomized trials are warranted to refine patient selection criteria, optimize dosing regimens, and evaluate long-term outcomes.

### 3.3. Role of the Transjugular Intrahepatic Portosystemic Shunt (TIPS): State of the Art

The transjugular intrahepatic portosystemic shunt (TIPS) represents a widely accepted intervention for addressing various complications (including both AREs and NAREs) related to portal hypertension in the dACLD setting [27,91]. In recent years, the role of TIPS has undergone significant evolution, marked by advancements in procedural techniques (with particular reference to the shift from uncovered to covered stent placement), refinement of prognostic tools, and enlargement of the spectrum of clinical indications [27,91,92].

Classically, TIPS consists of percutaneously creating a vascular channel between the portal and hepatic veins, positioning devices (i.e., stent) designed to decompress the portal venous system in patients with ACLD and portal hypertension [27,91,92].

The rationale for TIPS lies in the relative ability to reduce the portosystemic pressure gradient: by diverting blood flow away from the high-resistance cirrhotic liver, TIPS restores the effective arterial blood volume, suppresses neurohormonal activation (including the RAAs), and ultimately improves renal perfusion and sodium excretion [93].

In the light of this, TIPS, by pathophysiologically targeting both the hemodynamic and neurohumoral dACLD derangements [94], represents a potentially useful approach for RecA/RA and kidney dysfunction. Several randomized controlled trials (RCTs) evaluating TIPS placement with uncovered stents versus LVP combined with HA infusion for the management of RecA/RA confirmed the efficacy of TIPS in controlling ascites, with success rates ranging from 60% to 80% [95–100]. In the sole RCT employing covered stents, ascites resolution was achieved in 89% of patients in the TIPS group, compared to just 29% in the LVP plus HA group [96]. Additionally, an individual patient data meta-analysis involving 3949 individuals found that TIPS placement for ascites management was linked to a reduced cumulative risk of further hepatic decompensation events—including ascites recurrence, variceal hemorrhage, HE, jaundice, HRS-AKI, and PBS [91].

Anyway, whereas the adoption of TIPS in managing both RA and RecA that is not responsive to medical therapy constitutes a largely supported and recommended approach [27,91], the specific role of TIPS in patients with kidney dysfunction remains a current and much-debated topic, since limited data in the literature on this clinical application have been provided by outdated observations.

In a meta-analysis of nine studies, Song et al. reported a similar proportion of renal function improvement in HRS-AKI and any HRS phenotype (HRS-AKI: 93%; any HRS: 84%) for patients receiving TIPS, revealing, anyway, a modest improvement in survival after this intervention (1-year survival rate: 47% in HRS-AKI vs. 64% in HRS-NAKI) [101].

A measurable change in the estimated glomerular filtration rate (eGFR) typically emerges within 3 to 4 months following TIPS placement, with potential renal benefits observed initially, particularly in patients with chronic kidney disease (CKD) defined by a GFR below 60 mL/min [96,102,103]. This evidence suggested that TIPS may disrupt the progressive decline in renal function driven by a reduced effective circulating volume. Notably, more recent findings indicate that TIPS creation results in an early, marked, and sustained improvement in renal parameters among patients with HRS-CKD, regardless of their initial CKD stage [96,102,103].

However, despite these physiological gains, data on clinical outcomes remain limited for patients with more advanced renal impairment—such as those with serum creatinine levels exceeding 3 mg/dL—as these individuals were frequently excluded from clinical trials [95–98,104].

Furthermore, the role of TIPS in patients undergoing dialysis has not been systematically evaluated, with existing evidence restricted to isolated case reports [105]. Finally, no RCTs have addressed the use of TIPS in the context of HRS-NAKI.

Therefore, collectively considering the above-mentioned evidence, the recent EASL CPGs propose a distinction between HRS-AKI and HRS-NAKI recommendations [27].

Currently, in dACLD individuals developing ascites and HRS-AKI non-responders to HA with vasoconstrictor (terlipressin) combination therapy, TIPS cannot be recommended (or must be considered only in very selected cases), due to the severe liver failure (i.e., a contraindication to non-urgent TIPS) featuring most of these patients, with the LT remaining the best possible choice [27]. Contrariwise, despite the weakness of recommendations due to the lack of RCTs, TIPS may be considered in patients with cirrhosis, ascites, and HRS-NAKI to reduce morbidity and mortality [27].

Considering routine clinical practical implications, in these scenarios, patient selection for TIPS vs long-term HA in HRS-NAKI remains a critical challenge, and opting for either strategy should be guided by the underlying pathophysiology, reversibility of renal dysfunction, and LT candidacy.

On one side, TIPS appears primarily indicated in patients with RA and portal hypertension who maintain adequate hepatic function (Child–Pugh score $\leq$ B7–B9 without HE) and no significant cardiopulmonary contraindications.

Emerging data suggest that TIPS may improve renal perfusion and reverse HRS in selected patients by decreasing portal pressure and systemic inflammation, remaining contraindicated in ACLF (grade $\geq$ 2), active infection, and severe liver failure [27]. In contrast, long-term HA infusion, as demonstrated in the ANSWER trial, has been associated with improved survival, fewer episodes of HRS, and a reduced need for LVP in patients with RecA or RA and stable liver function, especially those not eligible for TIPS or LT [71].

Comprehensively, this evidence suggests TIPS as the preferable approach in patients with preserved hepatic reserve, proposing, contrariwise, the long-term HA administration as a more suitable strategy for non-transplant candidates with chronic ascites (RecA/RA) and low-grade renal dysfunction. Anyway, in this setting, a multidisciplinary (hepatological–nephrological) approach is required to properly select patients with kidney dysfunction eligible for this procedure, simultaneously considering the primary indication, individualized risk factors, and physiologic goals of this intervention

### 3.4. Role of Alphapump®: State of the Art

Not negligible adverse effects and routinely observed complications associated with invasive procedures (i.e., repeated LVPs or TIPS placement) [92,106], in addition to the limited availability of donor organs (restricting the LT chances) [107], have progressively given impulses to the development of innovative therapeutic approaches in the management of RecA and RA in dACLD [108].

In this setting, the automated low-flow ascites pump (Alfapump® Sequana Medical NV, Europe, Kortrijksesteenweg 1112, 9051 Sint-Denijs-Westrem, Belgium) has emerged as an encouraging strategy. Alfapump® is a subcutaneously implanted, battery-powered device designed to transfer ascitic fluid from the peritoneal cavity to the urinary bladder [109]. This system enables the controlled removal of ascites in terms of both timing and volume, allowing fluid to be excreted naturally through micturition [109]. In December 2024, this device received Food and Drug Administration (FDA) approval for the treatment of both RecA and RA in dACLD patients in the United States.

Anyway, this intervention is not currently approved by other regulatory authorities, including the European Medicines Agency (EMA) or Health Canada [109].

The first data on the safety and performance of the Alfapump® system were reported in the PIONEER study (2013), which enrolled 40 patients across 9 centers, with a follow-up period of 6 months post-implantation. During this time, the mean number of large-volume paracenteses (LVPs) decreased significantly from 3.4 to 0.2 per month.

However, approximately one-third of the devices required explantation, primarily due to infection, followed by catheter dislodgement [110]. Based on these findings, long-term antibiotic prophylaxis with norfloxacin was recommended for all patients carrying an Alfapump® system [110]. A few years later, in the first multicenter RCT by Bureau, Adebayo et al., the Alfapump® system was associated with a reduced need for LVPs and improved quality of life and nutritional parameters in dACLD patients, without reporting significant effects on survival [111]. Although the incidence of serious adverse events (mainly AKI) and hospitalizations was significantly higher in the Alfapump® group, these episodes were generally limited in severity and did not affect overall survival [111]. Solà et al. subsequently investigated the specific impact of the pump on the renal and circulatory function in a prospective study including 10 patients with liver cirrhosis and RA, followed up for 1 year, considering the variations in eGFR as the primary research outcome [112]. Notably, a significant worsening of eGFR (decreased from 67 mL/minute/1.73 $m^2$ at baseline to 45 mL/minute/1.73 $m^2$ at month 6) was observed, reporting AKI as the most frequent episode of complications in these patients [112].

Interestingly, this effect on GFR was associated with a marked increase in plasma renin activity and norepinephrine concentrations [112], and the use of HA has been proposed as a potential modulator of these effects, warranting further investigation [112].

In conclusion, Alfapump®, reducing the occurrence of LVPs, is a system that has advantages in improving the quality of life of patients, even if its benefit in survival is uncertain. Due to the non-infrequent rate of complications, even severe, patient selection and proof of indication before the implant are very crucial [109].

Anyway, despite the encouraging effects in the dACLD-RecA/RA setting, the deleterious reported repercussions on renal performance preclude, at the moment, the adoption of this strategy in the management of dACLD-ascites-associated kidney dysfunction [109,113].

*3.5. Managing Infections Properly: Crucial Impact on Ascites and Kidney Dysfunction Worsening*

Infections are among the most frequent and severe complications in dACLD patients, have a critical role in precipitating both acute kidney injury (AKI) and worsening ascites [114]. Bacterial infections, particularly SBP, urinary tract infections, and pneumonia, can lead to abrupt circulatory changes, systemic inflammation, and immune dysregulation, which collectively trigger or exacerbate organ dysfunction, including kidney dysfunction and renal failure, in the most critical stages of liver disease [114,115]. In this setting, indeed, infections often act as precipitating events for ACLF, through mechanisms involving systemic inflammation and cytokine storms, which impair renal perfusion and induce functional and structural kidney injury [115]. About this, in a large prospective cornerstone study, a significant proportion (up to 50%) of dACLD patients presenting with bacterial infections at admission developed ACLF, and, in this cohort, renal failure was the most reported organ dysfunction [28]. Relevantly, failure to resolve bacterial infection in dACLD patients with ascites was demonstrated to be a variable independently associated with the development of renal failure [116]. Furthermore, infections were significantly associated with reduced survival in patients with ascites, due to both the direct impact on hemodynamic stability and increased risk of complications, including particularly HRS-AKI and RecA/RA [3,28].

Therefore, considering this dramatic clinical scenario, effective management of infections in dACLD represents is a cornerstone of preventing or limiting the progression of both ascites and kidney dysfunction. Early diagnosis and prompt empirical antibiotic therapy are essential, as delayed treatment is associated with higher rates of AKI, ACLF, and mortality [3,117]. In support of this, in patients with SBP, the immediate initiation of third-generation cephalosporins or carbapenems (in high-risk or nosocomial cases) along

with proper i.v. HA administration has been revealed, and subsequently confirmed, to significantly reduce the incidence of HRS-AKI and, ultimately, improve survival [3,71,74].

However, despite these encouraging findings, the other side of the coin is represented by the increasing antimicrobial resistance in cirrhotic patients, currently posing new challenges [114]. Alarmingly, in this setting, multidrug-resistant organisms (MDROs) have been recently isolated in up to 30–50% of infections, especially in patients requiring frequent hospitalizations or previously exposed to antibiotic "pressure" [118]. In these cases, growing evidence has untied the Gordian knot of rapidly starting nonspecific treatment, revealing how an inappropriate empiric therapy fails to control infection, simultaneously promoting the deterioration of renal function and the uncontrolled ascites [119,120].

Regarding this, the large international EABCIR-Global study highlighted a relevant proportion (over 30%) of dACLD patients with bacterial infections who were nonresponders to empiric antibiotic therapy. Relevantly, these patients showed significantly higher rates of septic shock, organ failure—including AKI—and increased mortality [120].

Furthermore, in dACLD patients with bacteremia, inappropriate initial antimicrobial therapy was linked to significantly lower 30-day survival, emphasizing the prognostic weight of early and accurate empiric treatment [119].

Taken together, these findings underscore the need for early and targeted antibiotic strategies in this setting, remarking on the dramatic consequences derived from adopting inadequate initial treatment, both in terms of failure to control infection and the contribution to kidney dysfunction. In this sense, future research efforts should focus on the identification of strategies providing tailored risk stratification, rational antibiotic use, and the implementation of antimicrobial stewardship programs in clinical practice.

### 3.6. Managing Kidney Dysfunction in Acute-on-Chronic Liver Failure: A Hard Clinical Challenge

The proper management of HRS-AKI in patients developing ACLF, particularly at advanced stages, represents one of the most demanding clinical challenges in hepatology [3]. As previously discussed, ACLF constitutes a distinct and dynamic syndrome characterized by intense systemic inflammation, immune dysregulation, and rapid multiorgan deterioration [28]. These peculiar and critical features render standard strategies, including vasoconstrictors and HA administration, less effective in this setting [28].

Relevantly, in line with this, in a large multicenter cohort study, the probability of response to terlipressin and HA administration was revealed to proportionally diminish with an increasing ACLF grade [28]. Specifically, patients with an ACLF grade 3 had the lowest response rates, highlighting the limited efficacy of current pharmacological options in this subgroup [28]. These initial findings were later confirmed in the CONFIRM trial, where, dramatically, only 12% of patients with ACLF grade 3 responded to terlipressin and HA therapy [121]. This low response rate has been attributed to the severity of the systemic inflammatory response and marked systemic oxidative stress in ACLF grade 3, simultaneously contributing to the subclinical tubular damage at the renal level [42]. Consequently, unlike "classical" HRS-AKI, this form of injury appears less reversible by merely improving renal perfusion [122].

Furthermore, besides efficacy-related issues, safety concerns have been raised regarding the use of terlipressin in ACLF grade 3.

In the CONFIRM trial, patients presenting with ACLF grade 3 showed a relevant (up to 30%) incidence of respiratory failure events, which represented a major contributor to overall mortality, suggesting elevated caution level when prescribing terlipressin in this setting [121]. Importantly, although responders to terlipressin and HA administration in ACLF grade 3 showed a modest reduction in 28-day mortality, this did not translate into a significant improvement in 90-day survival (71% vs. 80% in responders vs. non-responders) [121].

This "timing-based" evidence highlights that short-term hemodynamic improvements may really benefit patients who are candidates for LT, appearing prognostically futile in those not eligible for LT.

Therefore, managing HRS-AKI in ACLF represents a hard clinical challenge, requiring a tailored approach that should not be limited to focusing on the potential for renal recovery and should comprehensively consider the overall prognosis, eligibility for transplantation, and risks associated with therapy. In this scenario, early referral for LT evaluation and inclusion in transplant programs should be prioritized in patients showing even minimal improvement, whereas conservative management may be more appropriate in those with no transplant options, also considering the emerging role of a palliative strategy in this setting [123].

### 3.7. The Emerging Role of Palliative Care (PC)

Despite advances in the management of dACLD-related complications and ACLF, a significant proportion of patients are not candidates for LT due to heterogeneous causes, including advanced age, comorbidities, frailty, or lack of access to transplant programs [124,125]. In such cases, palliative care (PC) becomes an essential component of patient-centered management, aiming to improve quality of life, relieve suffering, and support shared decision-making [123].

The clinical course of ACLF is often unpredictable, with episodes of rapid deterioration, multiorgan failure, and high short-term mortality—even in patients who initially present with moderate disease severity [28,117]. In these contexts, the early integration of PC can help align treatment goals with patient values and preferences, especially when curative or disease-modifying options are no longer feasible. About this, a recent systematic review underscored the pressing need for PC strategies in this population, highlighting how dACLD and ACLF patients often experience a high symptom burden—comparable to that reported in terminal cancer—yet remaining significantly under-referred to PC teams [126]. Symptoms such as pain, breathlessness, pruritus, confusion, ascites-related discomfort, and dysgeusia are frequently under-recognized and inadequately treated, leading to unnecessary hospitalizations and distress at the end of life [126,127]. Moreover, families and caregivers often face complex decisions about life-sustaining treatments, including mechanical ventilation, renal replacement treatment, and vasopressor support, frequently without prior goals-of-care discussions [123,126].

Conclusively, incorporating PC into standard hepatology care, particularly for patients with ACLF grade 3 or those not eligible for LT, can facilitate timely conversations around prognosis, advance care planning, and transitions to comfort-focused care. Strengthening these interventions may also improve satisfaction with care and reduce the use of non-beneficial interventions in the final days of life [123]. Future research should explore models for earlier PC integration in dACLD and ACLF management pathways, as well as strategies to overcome barriers to referral.

## 4. Future Perspectives

The research field of dACLD and portal hypertension is extensive and continuously evolving. In this scenario, in parallel with growing evidence elucidating the numerous underlying pathogenic mechanisms, emerging studies address the identification of novel therapeutic targets, continuously proposing modern potential applications in the management of dACLD patients developing ascites and kidney dysfunction.

In this sense, the research efforts have mainly focused on targeting gut-dysbiosis-induced systemic inflammation/immune dysfunction and the hemodynamic peritoneal-renal-related alterations promoting ascites recurrence and kidney dysfunction in dACLD,

investigating the potential role of fecal microbiota transplantation (FMT) and peritoneal dialysis (PD), respectively.

### 4.1. Fecal Microbiota Transplantation (FMT): Rationale of a Nascent Approach

As described above, in ACLD, the increased intestinal permeability and dysbiosis facilitate the intestinal BT, triggering a systemic inflammatory response [128,129]. This process contributes to the development of portal hypertension and the progression of liver disease itself, establishing a vicious cycle that further impairs the gut barrier [128,129].

Fecal microbiota transplantation (FMT) could interrupt the vicious cycle of dysbiosis-bacterial translocation–systemic inflammation by acting on multiple levels: restoring eubiosis, strengthening the intestinal barrier, and reducing systemic immune activation [130,131]. These effects could have an indirect positive impact on portal hypertension and the progression of renal dysfunction in ACLD.

Regarding the eubiosis restoration, recent studies suggest that FMT increases microbial diversity, favoring protective strains that compete with potentially pathogenic enterobacteria [130]. Moreover, "healthy" microbes from FMT produce butyrate and other short-chain fatty acids (SCFAs), which strengthen the intestinal tight junctions. This effect reduces intestinal permeability, ultimately limiting BT and reducing the production of inflammatory cytokines (TNF-$\alpha$, IL-6), with ameliorative repercussions on splanchnic vasodilation and renal function [132].

Although the efficacy of FMT in chronic non-infectious diseases (such as CKD or RA) remains to be validated, some preclinical evidence suggests that, in mouse models of diabetes and nephropathy, transplantation from healthy donors reduced albuminuria and intestinal inflammation [133].

However, to date, the number of RCTs evaluating FMT in chronic liver disease remains limited, and further investigations in humans are needed. Therefore, in this setting, although preclinical data are encouraging, the clinical application of FMT in chronic liver disease remains in its infancy. The current body of evidence is constrained by small sample sizes, heterogeneous patient populations, short follow-up durations, and limited safety data. Furthermore, potential complications, including the infection risk and immunologic reactions, warrant careful consideration. To advance the translational potential of FMT in ACLD, future research should prioritize large-scale randomized controlled trials with standardized protocols, extended follow-up, and rigorous safety monitoring. Looking at practical clinical implications, FMT may find a realistic role in selected cases of RA or recurrent infections (particularly SBP), both representing scenarios where dysbiosis plays a pivotal pathogenic role. Moreover, by potentially mitigating the inflammatory milieu that precipitates HRS-AKI and ACLF, FMT could evolve into an adjunctive therapy to delay decompensation or enhance the response to HA-based strategies. However, larger randomized trials are needed before routine clinical use, and their application remains limited to specialized centers with strict donor-screening protocols.

In this sense, a clearer understanding of patient selection criteria, optimal donor profiles, and mechanistic endpoints will be essential to validate FMT as a viable adjunctive therapy in the management of portal hypertension and renal dysfunction in ACLD.

### 4.2. Peritoneal Dialysis (PD): Rationale and Potential Clinical Applications

Peritoneal dialysis (PD) represents a valid strategy in the management of patients with cirrhotic ascites and concomitant end-stage kidney disease (ESKD). PD is a form of renal replacement therapy (RRT) that utilizes the peritoneal membrane of the patients as a semipermeable filter to facilitate the removal of metabolic waste products and excess fluid from the bloodstream [134]. This technique serves as an alternative to hemodialysis

(HD). In particular, contrariwise to hemodynamic (i.e., hypotension) and hemorrhagic complications characterizing HD, peritoneal dialysis (PD) represents a valid alternative, providing hemodynamic stability, obviating the need for anticoagulant drugs, and offering regular drainage of ascites in RA-affected individuals [134].

PD consists of the infusion of a sterile dialysis solution into the peritoneal cavity, where solute and fluid exchange occur across the peritoneal membrane via diffusion and osmosis. The spent dialysate is then drained and replaced cyclically [134]. PD can be performed manually through Continuous Ambulatory Peritoneal Dialysis (CAPD) or with the aid of an automated device in Automated Peritoneal Dialysis (APD) [134,135].

Clinical evidence indicates that PD, compared to hemodialysis, offers hemodynamic stability, the continuous drainage of ascites, and a reduced need for repeated LVP, with overall mortality similar to or lower than that observed with HD [135].

Although the risk of peritonitis is slightly increased in cirrhotic patients, rates of mechanical complications (hernias, fluid leaks, protein losses) are comparable or manageable in the long term, with initial protein losses tending to stabilize over time [135].

This evidence suggests that PD may represent a well-tolerated therapeutic option in patients with RA and advanced kidney disease, offering simultaneously RRT and the assisted control of ascites, avoiding invasive strategies (repeated LVP or TIPS placement). However, the standardization of indications, catheter insertion protocols, nutritional balancing, and strategies to minimize protein losses remains necessary, representing the main challenge of the research field addressing this topic [134].

Importantly, the current evidence base is constrained by several factors: small and heterogeneous patient cohorts, limited follow-up durations, and a lack of standardized protocols. These limitations hinder the generalizability of findings and underscore the need for future large-scale, multicenter clinical trials [134]. Further research should aim to define optimal patient selection criteria, refine procedural techniques, and establish long-term safety and efficacy profiles. Conclusively, while PD offers a promising dual benefit—renal support and ascites control—in patients with dACLD and worsening ascites, its broader adoption will depend on rigorous validation through well-designed clinical research and the development of standardized care pathways.

### 4.3. Punctual Risk Stratification and Accurate Prediction of Outcomes: Is It Time to Go Beyond "General" Phenotype Classification and Consider "Tailored" Approaches?

Early identification and risk stratification in patients with dACLD complicated by ascites and kidney dysfunction are critical to improving outcomes and preventing irreversible organ damage [136]. As recently highlighted by Carrier et al., RA marks a pivotal turning point in the natural history of ACLD, predisposing one to kidney dysfunction, and HRS-AKI represents an ARE that often develops silently in this setting, making early diagnostic vigilance essential [136].

For this purpose, a decade ago, Angeli et al. already emphasized that even mild elevations in serum creatinine can predict AKI and are strongly associated with an increased in-hospital mortality [137].

This cornerstone revolutionary concept proposed the abandonment of a static serum creatinine cutoff (e.g., >1.5 mg/dL) to identify renal impairment, and remarking on dynamic changes in creatinine levels can be clinically significant in dACLD patients [137].

Receiving this precious evidence, the recent ADQI-ICA joint consensus meeting recommended the dynamic monitoring of renal function and early-stage AKI recognition, proposing—as previously described—novel diagnostic criteria for HRS-AKI [26].

Emerging research has evaluated the applications of the ADQI-ICA proposed new criteria in real life, focusing on therapeutic strategies and relevant prognostic repercussions, with particular reference to the mortality outcome. A recent observation of a large cohort

(374 patients) of cirrhotic patients developing kidney dysfunction explored how classifying acute kidney injury (AKI) affects mortality outcomes [138]. In this study, HRS was identified as the HRS-AKI in 17.4%, while 82.6% were classified as having HRS-NAKI, with 62.6% specifically diagnosed with ATN [138].

Unexpectedly, although a higher mortality rate was reported in HRS-AKI, HRS-AKI was not associated with a significantly higher 90-day mortality risk in comparison with HRS-NAKI (ATN) [138]. Interestingly, besides vasopressor administration and transplant listing status, RRT emerged as a significantly predictive variable, rather than an HRS phenotype [138]. Moreover, in a recent multicenter cohort study of hospitalized patients with AKI requiring renal replacement therapy (AKI-RRT), the etiology of AKI, whether HRS-AKI or HRS-NAKI, was not independently associated with 90-day mortality [139]. This observation challenges the traditional view that HRS-AKI, often considered the most severe form of AKI in dACLD, carries a distinctly worse prognosis.

Revolutionarily, these findings suggest that, contrary to the general non-cirrhotic inhomogeneous population where RRT has been associated with poor outcomes [140], in cirrhotic patients, once RRT is initiated, the underlying cause of AKI may be less prognostically relevant than previously thought, in a scenario where, anyway, the global epidemiology of AKI-RRT in hospitalized ACLD patients remains unknown [139].

Therefore, in this context, dACLD-RA patients should be approached not merely according to the diagnostic-criteria-configured kidney dysfunction subtype (HRS-AKI or HRS-NAKI), moving beyond "general" phenotype classification and considering the role of "tailored" management strategies in impacting these outcomes, primarily based on a personalized risk stratification [141].

This paradigm shift is echoed in evidence, remarking on the absolute need to integrate non-invasive tools (NITs), including elastography and Fibrosis-4 (FIB-4) scoring, to properly stratify risk and guide personalized management strategies [141–143].

In this sense, the combination of available NITs has emerged to offer the more effective prediction of prognosis and stratification of the risk of mortality in comparison to the adoption of a single tool (e.g., MELD alone or Child–Pugh score alone determination) [143,144], which can potentially fall short in capturing the full physiopathological (hemodynamic and immune-related) complexity of patients with both ascites and renal dysfunction.

About this, Hasan et al., by combining MELD, Child–Pugh, and leukocyte levels, developed a scoring system presenting an extraordinary performance (AUC of 0.921) in predicting 90-day mortality in hospitalized cirrhotic patients, accurately stratifying patients into low-, moderate-, and high-risk categories [145]. More recently, novel Artificial Intelligence (AI)–machine learning (ML) models were developed to predict mortality in dACLD. These models consistently outperformed traditional scores (with AUROC values ranging from 0.71 to 0.96), suggesting that integrating broader clinical and demographic data can significantly improve the prognostic accuracy [146].

Besides this, the inclusion of body composition data with the integrated assessment of nutritional status represents another crucial moment in the designation of tailored management strategies based on the personalized risk stratification of dACLD patients with RA and kidney dysfunction [147].

In particular, in dACLD patients with worsening ascites and renal impairment, the persistent ascites-induced abdominal distension, simultaneously with other hepatologic-related and uremic-associated physiopathological drivers contributing to dysgeusia (i.e., altered taste perception), determines decreased appetite levels and reduced dietary intake, promoting malnutrition and sarcopenia occurrence [127]. Sarcopenia, in turn, reflects the poor nutritional status and contributes to worse outcomes, including an increased

risk of AREs/NAREs, hospitalizations, and mortality [147]. This evidence highlights the importance of early nutritional assessments and intervention, including addressing taste alterations, in the management of patients with advanced liver disease [147].

Altogether, these findings support the relevance of a multidisciplinary, proactive, and tailored approach, replacing the outdated "one size fits all" model with personalized care based on early and accurate risk profiling of dACLD patients developing ascites and kidney dysfunction [148]. However, current studies are limited by small sample sizes, short follow-up periods, and heterogeneous patient populations. The standardization of diagnostic criteria, longitudinal safety data, and large-scale prospective trials are urgently needed to validate these promising approaches and ensure their safe and effective integration into clinical practice [149–151].

## 5. Conclusions

Renal dysfunction in dACLD patients with ascites represents a common plague and may manifest as various phenotypes, with distinct and progressively worsening prognostic implications [152]. The coexistence of liver disease and renal impairment—whether acute or chronic—defines a high-risk clinical challenge, demanding intensive surveillance and individualized therapeutic approaches, configuring a scenario where, despite recent progress, the prognosis of these patients remains poor.

In the era of personalized medicine, there is an urgent need for tailored strategies to concretely improve the outcomes of this population. For this purpose, an enhanced understanding of the pathophysiological mechanisms driving these conditions appears to be crucial. Future studies should focus on bridging the existing knowledge gaps and translating the emerging evidence into clinical practice, ultimately fostering more nuanced and impactful patient care.

**Author Contributions:** M.D., M.R. and F.D.N.: guarantor of the article, conceptualization, methodology, investigation, and writing the original draft; P.V. and C.N.: conceptualization, methodology, formal analysis, investigation, and writing the original draft. S.B., C.G. and L.D.N.: investigation, resources, data curation, and visualization; A.F.: conceptualization, data curation, supervision. All authors have read and agreed to the published version of the manuscript.

**Funding:** This research received no external funding.

**Institutional Review Board Statement:** Not applicable.

**Informed Consent Statement:** Not applicable.

**Data Availability Statement:** No new data were created or analyzed in this study. Data sharing is not applicable to this article.

**Conflicts of Interest:** The authors declare no conflicts of interest.

## Abbreviations

The following abbreviations are used in this manuscript:

| | |
|---|---|
| ACLD | Advanced Chronic Liver Disease |
| ADH | Antidiuretic Hormone |
| ADQI | Acute Disease Quality Initiative |
| AKD | Acute Kidney Disease |
| AKI | Acute Kidney Injury |
| ALD | Alcoholic Liver Disease |
| APD | Automated Peritoneal Dialysis |
| AREs | Ascites-Related Events |
| ASC | Ascites |

| | |
|---|---|
| ATN | Acute Tubular Necrosis |
| CAID | Cirrhosis-Associated Immune Dysfunction |
| CAPD | Continuous Ambulatory Peritoneal Dialysis |
| CPGs | Clinical Practice Guidelines |
| CRP | C-Reactive Protein |
| CKD | Chronic Kidney Disease |
| dACLD | Decompensated Advanced Chronic Liver Disease |
| EASL | European Association for the Study of the Liver |
| EMA | European Medicines Agency |
| eGFR | Estimated Glomerular Filtration Rate |
| FIB-4 | Fibrosis-4 Score |
| FMT | Fecal Microbiota Transplantation |
| HA | Human Albumin |
| HE | Hepatic Encephalopathy |
| HPA | Hypothalamic–Pituitary–Adrenal |
| HRS | Hepatorenal Syndrome |
| HRS-AKI | Hepatorenal Syndrome–Acute Kidney Injury |
| HRS-AKD | Hepatorenal Syndrome–Acute Kidney Disease |
| HRS-CKD | Hepatorenal Syndrome–Chronic Kidney Disease |
| HRS-NAKI | Hepatorenal Syndrome–Non-Acute Kidney Injury |
| HD | Hemodialysis |
| IL | Interleukin |
| ICA | International Club of Ascites |
| LPS | Lipopolysaccharide |
| LRE | Liver-Related Event |
| LT | Liver transplant/Liver transplantation |
| LVP | Large-Volume Paracentesis |
| MELD | Model for End-Stage Liver Disease |
| MDROs | Multidrug-resistant organisms |
| ML | Machine Learning |
| NAKI | Non-Acute Kidney Injury |
| NAREs | Non-Ascites-Related Events |
| NITs | Non-Invasive Tools |
| NSBBs | Non-Selective Beta-Blockers |
| PAMPs | Pathogen-Associated Molecular Patterns |
| PD | Peritoneal Dialysis |
| PRRs | Pattern Recognition Receptors |
| RA | Refractory Ascites |
| RAAS | Renin-Angiotensin-Aldosterone System |
| RecA | Recurrent Ascites |
| RRT | Renal Replacement Therapy |
| SBP | Spontaneous Bacterial Peritonitis |
| SCFAs | Short-Chain Fatty Acids |
| SNS | Sympathetic Nervous System |
| TIPS | Transjugular Intrahepatic Portosystemic Shunt |
| TNF-$\alpha$ | Tumor Necrosis Factor-alpha |
| TLRs | Toll-Like Receptors |

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
