# Peer review of "Managing Ascites and Kidney Dysfunction in Decompensated Advanced Chronic Liver Disease: From “One Size Fits All” to a Multidisciplinary-Tailored Approach"

_livers, doi:10.3390/livers5030046_

Round 1

Reviewer 1 Report

Comments and Suggestions for Authors

The current review aim to surmmary the new evidence regarding the management of cirrhotic patients with compliated ascites. Good job. There are several comments as follow.
1. In the title and asbstract, authors suggested the target population of the current paper is chornic advanced liver disease. However, the maintext is foucs on the patients with liver cirrhosis. Liver cirrhosis is a common advanced liver disease, but liver cirrhosis didn't equal to chornic advanced liver disease(i.e., ACLF).
2. For the management of liver cirrhosis, an international position statement (PMID: 37456673) comprehensively summarizes the important role of human albumin in liver cirrhosis and its complications (including HRS ).
3. Recently studies suggested thet terlipressin could potential benefit to the renal function cirrhotic patients with liver cirrhosis and ascites.

minor:
Figure 1. Among the abbrevations, ADR: adverse-reaction drugs, but in the figure is ARDs, Please correct it.
Figure 2. The abbrevation of clinically significant portal hypertension should consistent. CSPH or CS(PH) in Figure.

Author Response

Reviewer 1

The current review aim to surmmary the new evidence regarding the management of cirrhotic patients with complicated ascites. Good job. There are several comments as follow.

  1. In the title and abstract, the authors suggested the target population of the current paper is chronic advanced liver disease. However, the main text is foucs on the patients with liver cirrhosis. Liver cirrhosis is a common advanced liver disease, but liver cirrhosis didn't equal to chronic advanced liver disease (i.e., ACLF).

Reply: We sincerely thank the Reviewer for this precious comment. We are aware that the term “advanced chronic liver disease” (compensated “cACLD” and decompensated “dACLD”) is not equal to “liver cirrhosis” (compensated liver cirrhosis and decompensated liver cirrhosis) whose definition remains histological-based, as well as, we know that “ACLF” (“Acute on Chronic Liver Failure) represents another nosological entity configurated by defined criteria which can complicate the course, especially of decompensated patients. According to the novel concept reported by the Baveno VII consensus, indeed, […] <<the term “compensated advanced chronic liver disease (cACLD)” had been proposed to reflect the continuum of severe fibrosis and cirrhosis in patients with ongoing chronic liver disease. A pragmatic definition of cACLD based on liver stiffness measurement (LSM) is aimed at stratifying the risk of CSPH and decompensation at the point of care, irrespective of histological stage or the ability of LSM to identify these stages. Currently, both terms “cACLD” and “compensated cirrhosis” are acceptable, but not equal […] >>.

 Therefore, since all evidence and findings included in our research are consistent and updated with this novel concept, following the Reviewers’ suggestion, in the resubmitted version of our manuscript, we have standardized the definition of advanced chronic liver disease (cACLD and dACLD) throughout the entire text, making it consistent with the title and the abstract.

  1. For the management of liver cirrhosis, an international position statement (PMID: 37456673) comprehensively summarizes the important role of human albumin in liver cirrhosis and its complications (including HRS).

Reply: We sincerely thank the Reviewer for this valuable suggestion. In the resubmitted version of our manuscript, the recent international position statement was properly enclosed and discussed in the paragraph dedicated to the role of Human Albumin Administration in the clinical routine practice, proposing the state of the art.

  1. Recently studies suggested that terlipressin could potential benefit to the renal function cirrhotic patients with liver cirrhosis and ascites.

Reply: We sincerely thank the Reviewer for this precious comment. As suggested, in the resubmitted version of our manuscript, a paragraph discussing the role of terlipressin in the management of patients with decompensated advanced chronic liver disease and ascites has been properly added, focusing on the emerging research supporting the potential benefits deriving from the potential continuous administration of this drug, beyond the acute setting of HRS-AKI.

Minor:

  • Figure 1. Among the abbrevations, ADR: adverse-reaction drugs, but in the figure is ARDs, Please correct it.
  • Figure 2. The abbrevation of clinically significant portal hypertension should consistent. CSPH or CS(PH) in Figure.

Reply: We thank the Reviewer for this notification. In the resubmitted version of our manuscript, all the notified abbreviations of both the Figures (Figure 1 and Figure 2) have been properly and consistently modified.

Reviewer 2 Report

Comments and Suggestions for Authors

This review explores the interrelationship between complicated ascites and renal dysfunction in patients with decompensated advanced chronic liver disease (dACLD). The authors synthesize current evidence on mechanisms beyond the traditional vasodilation hypothesis, highlighting roles for systemic inflammation, gut dysbiosis, and cirrhosis-associated immune dysfunction. They also summarize therapeutic options ranging from albumin infusions to TIPS, Alfapump®, and emerging strategies such as fecal microbiota transplantation.

This is a very comprehensive and timely review on a clinically relevant topic, written with good structure and clarity. However, I believe two important conceptual points are missing. First, the paper largely frames the discussion in the binary of compensated chronic liver disease versus decompensated chronic liver disease, but omits acute-on-chronic liver failure (ACLF) as a distinct and clinically critical step in this continuum. ACLF is increasingly recognized as a syndrome of intense systemic inflammation and multi-organ failure, which represents the “bridge” between cCLD and dCLD, and its absence here leaves a significant conceptual gap. Please refer to the latest relevant references.  https://www.wjgnet.com/1948-5182/full/v14/i12/2025.htm   

Second, the pathophysiology section could be strengthened by incorporating the paradigm of high-output cardiac failure as an underappreciated driver of renal injury in cirrhosis. While vasodilation and systemic inflammation are well described, the circulatory strain of hyperdynamic cardiac output and its contribution to renal hypoperfusion merit attention, as these mechanisms have therapeutic implications. Please refer to the latest relevant references. https://www.sciencedirect.com/science/article/abs/pii/S030698771200151X .

The abstract and results are otherwise well presented, but the writing occasionally drifts into overly general phrasing (e.g., “tailored approaches are needed” appears without concrete examples). It would be more useful if the authors specified how these approaches might be operationalized—such as patient selection for TIPS versus long-term albumin infusion, or where FMT may realistically enter clinical practice. Tables and figures are well structured, but I recommend adding a schematic figure that integrates the classical vasodilation model with newer concepts (systemic inflammation, CAID, dysbiosis, cardiac dysfunction, ACLF).

In addition, the review would benefit from explicitly addressing two neglected but clinically essential aspects: infection and palliative care. Infection is a leading trigger and complication in patients with ascites and kidney dysfunction, particularly spontaneous bacterial peritonitis, and should be discussed as part of the management continuum. Likewise, palliative care is increasingly recognized as an integral component of advanced liver disease management, particularly in those with refractory ascites and renal failure where transplant is not an option. Integrating these perspectives would make the review more holistic and clinically relevant. Please refer to the latest relevant references. https://www.wjgnet.com/2222-0682/full/v14/i4/95904.htm  .

Finally, while references are current, some seminal ACLF literature from the EASL-CLIF consortium should be cited, given its centrality to the field. With these additions, the review would offer a more balanced and modern view of the pathophysiological spectrum. Please refer to the latest relevant references. https://linkinghub.elsevier.com/retrieve/pii/S0016508513002916 .

This is a strong review, but needs incorporation of ACLF as a disease stage and acknowledgment of high-output cardiac failure in renal injury, plus sharper examples of tailored management strategies.

Author Response

Reviewer 2

This review explores the interrelationship between complicated ascites and renal dysfunction in patients with decompensated advanced chronic liver disease (dACLD). The authors synthesize current evidence on mechanisms beyond the traditional vasodilation hypothesis, highlighting roles for systemic inflammation, gut dysbiosis, and cirrhosis-associated immune dysfunction. They also summarize therapeutic options ranging from albumin infusions to TIPS, Alfapump®, and emerging strategies such as fecal microbiota transplantation.

This is a very comprehensive and timely review on a clinically relevant topic, written with good structure and clarity. However, I believe two important conceptual points are missing.

  1. First, the paper largely frames the discussion in the binary of compensated chronic liver disease versus decompensated chronic liver disease, but omits acute-on-chronic liver failure (ACLF) as a distinct and clinically critical step in this continuum. ACLF is increasingly recognized as a syndrome of intense systemic inflammation and multi-organ failure, which represents the “bridge” between caCLD and daCLD, and its absence here leaves a significant conceptual gap. Please refer to the latest relevant references. https://www.wjgnet.com/1948-5182/full/v14/i12/2025.htm  

Reply: We sincerely thank the Reviewer for this precious comment. In the resubmitted version of our manuscript, by also referring to the latest relevant suggested references, acute-on-chronic liver failure (ACLF) has been properly integrated and discussed. In particular, a dedicated paragraph has been proposed to explore the relationship between kidney dysfunction, ascites, and ACLF, and passages have been completely dedicated to the hard clinical challenge represented by the treatment of renal failure in patients developing ACLF.

  1. Second, the pathophysiology section could be strengthened by incorporating the paradigm of high-output cardiac failure as an underappreciated driver of renal injury in cirrhosis. While vasodilation and systemic inflammation are well described, the circulatory strain of hyperdynamic cardiac output and its contribution to renal hypoperfusion merit attention, as these mechanisms have therapeutic implications. Please refer to the latest relevant references. https://www.sciencedirect.com/science/article/abs/pii/S030698771200151X

Reply:  We sincerely thank the Reviewer for this precious suggestion. According to this, in the revised version of our manuscript, the cardiac dysfunction with the paradigm of high-output cardiac failure has been properly integrated and discussed in the paragraph dedicated to the physiopathology, also referring to the last proposed relevant references. 

 Others:

The abstract and results are otherwise well presented, but the writing occasionally drifts into overly general phrasing (e.g., “tailored approaches are needed” appears without concrete examples). It would be more useful if the authors specified how these approaches might be operationalized—such as patient selection for TIPS versus long-term albumin infusion, or where FMT may realistically enter clinical practice.

Reply: We sincerely thank the Reviewer for this precious suggestion. In the revised version of our manuscript, general phrasing has been avoided, and therapeutic approaches (validated or potential), including patient selection for TIPS versus long-term albumin infusion and FMT future applications, have been properly and critically presented, discussing the concrete applications in the routine clinical practice

Tables and figures are well structured, but I recommend adding a schematic figure that integrates the classical vasodilation model with newer concepts (systemic inflammation, CAID, dysbiosis, cardiac dysfunction, ACLF).

Reply: We sincerely thank the Reviewer for this precious comment. As suggested, a schematic figure integrating the classical vasodilation model with newer theories has been added.

In addition, the review would benefit from explicitly addressing two neglected but clinically essential aspects: infection and palliative care. Infection is a leading trigger and complication in patients with ascites and kidney dysfunction, particularly spontaneous bacterial peritonitis, and should be discussed as part of the management continuum. Likewise, palliative care is increasingly recognized as an integral component of advanced liver disease management, particularly in those with refractory ascites and renal failure where transplant is not an option. Integrating these perspectives would make the review more holistic and clinically relevant. Please refer to the latest relevant references. https://www.wjgnet.com/2222-0682/full/v14/i4/95904.htm.

Reply: We thank the Reviewer for these valuable comments. As suggested, the crucial role of infections has been integrated in the revised version of our manuscript, and a paragraph dedicated to the relevance of managing these events has been added. Finally, as recommended, the role of palliative care has also been properly discussed.

Finally, while references are current, some seminal ACLF literature from the EASL-CLIF consortium should be cited, given its centrality to the field. With these additions, the review would offer a more balanced and modern view of the pathophysiological spectrum. Please refer to the latest relevant references. https://linkinghub.elsevier.com/retrieve/pii/S0016508513002916.

Reply: We thank the Reviewer for this suggestion. In the revised version of our manuscript, seminal ACLF literature from the EASL-CLIF consortium has been cited by considering the last relevant proposed references. Thanks to this, a more balanced and updated view of the pathophysiological spectrum has also been presented.

Conclusion: This is a strong review, but needs incorporation of ACLF as a disease stage and acknowledgment of high-output cardiac failure in renal injury, plus sharper examples of tailored management strategies

Reply: We thank the Reviewer for all these precious comments. As previously reported, and as suggested, the resubmitted and revised version of our manuscript incorporates ACLF, considering the role of high-output cardiac failure in renal injury, as well as providing sharper examples of tailored management strategies

Reviewer 3 Report

Comments and Suggestions for Authors

1. Terminology: Is "Complicated Ascites" a Standardized Term?
While the manuscript uses the term “complicated ascites (CA),” it is unclear to what extent this expression has been standardized in recent guidelines or the broader literature. 

2. Definitions: HRS-NAKI vs. HRS-AKD, and the 24-Hour Albumin Challenge
The manuscript references the concept of "HRS-NAKI" and other terminology about AKI, CKD, HRS... However, it is my understanding that the EASL guidelines and international consensus are now increasingly adopting the term HRS-AKD (Acute Kidney Disease). Is “HRS-NAKI” a well-accepted term in contemporary practice, or should the newer terminology (“HRS-AKD”) be used for clarity and alignment with guidelines?  Recently, terms such as HRS-AKD have been introduced, but the management of AKI is still largely based on evolving guidelines, and a unified, standardized approach has yet to be established, leading to ongoing uncertainty.

Additionally, reThe manuscript discusses the reduction of the albumin challenge to 24 hours, but there is considerable debate in the field about possible risks of prematurely shortening this assessment period. Some recent studies and expert opinions caution against reducing the timeframe to 24 hours, suggesting it may be too early to reliably differentiate HRS-AKI from other causes of AKI. Could the authors elaborate on their position regarding the 24-hour protocol versus the traditional 48-hour approach, especially in light of these controversies?

In my opinion, especially in EASL group, there is currently considerable debate regarding AKI, HRS and related conditions.

3. Pathophysiology: Why Is Ascites Worsening, Not Varices, with Chronic Inflammation?
The shift toward viewing chronic inflammation (and the gut-liver axis) as major drivers of hepatic decompensation is highly relevant and reflects current trends. However, the manuscript does not clearly explain why, in clinical practice, some patients experience worsening of ascites while others may see regression of varices. If chronic inflammation acts as a global driver of portal hypertension and multi-organ dysfunction, what is the mechanistic explanation for this clinical divergence—namely, the increased prevalence of severe/refractory ascites while the incidence of variceal complications seems to decrease? Further elaboration on this mechanistic aspect would greatly enhance the translational value of the review.

4. it is commendable that the authors address novel therapeutic and prognostic strategies—such as faecal microbiota transplantation (FMT), artificial intelligence (AI)-based risk prediction, peritoneal dialysis (PD), and the Alphapump device—with a clear clinical focus. These sections not only reflect the current research priorities but also underscore the rapidly evolving landscape of multidisciplinary care for this complex patient group. However, these interventions still lack robust long-term data and results from large-scale human randomized trials, so conclusions about their efficacy and safety should be approached cautiously. Highlighting the current limitations (e.g., restricted patient populations, short follow-up, risk of complications) and clearly stating the need for future large clinical studies and extended safety data would help provide a more nuanced and critical perspective on these promising approaches.

Author Response

Reviewer 3

  1. Terminology: Is "Complicated Ascites" a Standardized Term?

While the manuscript uses the term “complicated ascites (CA),” it is unclear to what extent this expression has been standardized in recent guidelines or the broader literature.

Reply: We sincerely thank the Reviewer for this relevant notification. In the initial version of our manuscript, we proposed the term “complicated” as the complement of the well-recognized and standardized “uncomplicated” ascites. Anyway, as brilliantly highlighted by the Review, the first has not yet been standardized, and it has never been reported by the clinical practice guidelines.

Therefore, according to the precious Reviewer’s suggestion, in the resubmitted version of our paper, this not-standardized term has been replaced (by the approved and well-recognized “Recurrent Ascites” and “Refractory Ascites” definitions), and thus properly avoided (this, in the title, in the abstract, in the main text, and in the figures with relative captions)

  1. Definitions: HRS-NAKI vs. HRS-AKD, and the 24-Hour Albumin Challenge
  • The manuscript references the concept of "HRS-NAKI" and other terminology about AKI, CKD, HRS... However, it is my understanding that the EASL guidelines and international consensus are now increasingly adopting the term HRS-AKD (Acute Kidney Disease). Is “HRS-NAKI” a well-accepted term in contemporary practice, or should the newer terminology (“HRS-AKD”) be used for clarity and alignment with guidelines? Recently, terms such as HRS-AKD have been introduced, but the management of AKI is still largely based on evolving guidelines, and a unified, standardized approach has yet to be established, leading to ongoing uncertainty.

Reply: We sincerely thank the Reviewer for this precious consideration, focusing on kidney dysfunction definitions in cirrhosis.

As consistently added and discussed also in the revised version of the paper (in the paragraph specifically discussing the kidney dysfunction phenotypes), according to the most recent “Acute Disease Quality Initiative (ADQI) and International Club of Ascites (ICA) joint multidisciplinary consensus on the topic” (PMID: 38527522), Acute Kidney Injury (AKI), Acute Kidney Disease (AKD), and Chronic Kidney Disease (CKD) have to be classified by Kidney Disease Global Outcome (KDIGO) criteria based on the duration and severity of structural and functional abnormalities.

Despite this classification, “[…] AKI, AKD, and CKD create a continuum whereby initial kidney injury can lead to recovery (adaptive repair), persistent renal injury, and/or eventually CKD (maladaptive repair). AKI is a subset of AKD; therefore, all patients with AKI are considered to have AKD. Multiple episodes of AKI may occur over the course of an illness within one individual. After AKI resolves, patients may still have abnormalities in kidney function and/or structure that fulfill the criteria for AKD. Patients with HRS-AKD meeting AKI criteria are classified as having HRS-AKI. HRS for less than 90 days would be classified as HRS-AKD, while HRS persisting for more than 90 days would be classified as HRS-CKD […]”. Anyway, in the same consensus (2024), there is no recommendation to definitely abandon the old (but, maybe gold) definition and termination “HSR-AKI”, as well as in more recent (2025) EASL clinical practice guidelines (including the recently published EASL Clinical Practice Guidelines on TIPS, PMID: 40180845), all the therapeutic approaches for ascites and kidney dysfunction continue to refer to the dualism “HRS-AKI”-“HRS-NAKI”, remarking on the fact that the management of AKI is still largely based on evolving guidelines, and a unified, standardized approach has yet to be established, leading to ongoing uncertainty.

  Therefore, consistent with this presented scenario, in the revised version of our manuscript, all these features and novelties have been properly discussed, as well as the novel concept of HRS-AKD has been properly presented in the paragraph dedicated to phenotypes presentation (where also Figure 1 has been uptaded), while preserving the “old” terminology in reporting the treatment strategies (i.e., TIPS), with the relative supporting evidence.

  • Additionally, the manuscript discusses the reduction of the albumin challenge to 24 hours, but there is considerable debate in the field about possible risks of prematurely shortening this assessment period. Some recent studies and expert opinions caution against reducing the timeframe to 24 hours, suggesting it may be too early to reliably differentiate HRS-AKI from other causes of AKI. Could the authors elaborate on their position regarding the 24-hour protocol versus the traditional 48-hour approach, especially in light of these controversies? In my opinion, especially in EASL group, there is currently considerable debate regarding AKI, HRS and related conditions.

Reply: We sincerely thank the Reviewer for this comment on this relevant point presented in our review, and we completely agree with His/Her point of view, remarking on the considerable and heated debate, particularly in the EASL group, concerning the management of kidney dysfunction/AKI-related conditions in the dACLD setting. This continuous “dynamic” scenario appears to be attributable to constantly growing evidence and cumulating emerging findings on this topic, disproving “solid theories” (based on logical rationale) that appeared as "certainties" even just a year before.

In this scenario, also considering the relevance (including precise experimental design with an adequate number of enrolled and, subsequently, followed patients) of very recent research showing that shortening the duration of albumin therapy may lead to overdiagnosis and overtreatment with terlipressin, we feel confident to agree with the position of Angeli et al. on this topic, retaining that following the EASL-AKI algorithm the (i.e., the traditional 48-hour approach) remains the best choice to manage HRS-AKI, at least at the moment. The last clarification appears obligatory, considering the constant evolution of knowledge in this field.

  1. Pathophysiology: Why Is Ascites Worsening, Not Varices, with Chronic Inflammation?

The shift toward viewing chronic inflammation (and the gut-liver axis) as major drivers of hepatic decompensation is highly relevant and reflects current trends. However, the manuscript does not clearly explain why, in clinical practice, some patients experience worsening of ascites while others may see regression of varices. If chronic inflammation acts as a global driver of portal hypertension and multi-organ dysfunction, what is the mechanistic explanation for this clinical divergence—namely, the increased prevalence of severe/refractory ascites while the incidence of variceal complications seems to decrease? Further elaboration on this mechanistic aspect would greatly enhance the translational value of the review.

Reply: We sincerely thank the Reviewer for this precious suggestion, and we firmly believe His/Her comment would decisively increase the translational potential of the present review.

According to this, in the resubmitted version of our manuscript, a paragraph dedicated to properly discussing the potential mechanisms (enclosing, among the other overviewed features, the intriguing view of inflammation as a “compartment-specific driver” and the consideration of a “district-specific” vascular remodeling) sustaining the dualism “Ascites worsening vs Varices stabilization/regression” has been added, simultaneously considering the potential prognostic and therapeutic repercussions of further elucidation of these pathogenetic events with relatively clinical applications in the routine management of dACLD patients. 

  1. It is commendable that the authors address novel therapeutic and prognostic strategies—such as faecal microbiota transplantation (FMT), artificial intelligence (AI)-based risk prediction, peritoneal dialysis (PD), and the Alphapump device—with a clear clinical focus.

These sections not only reflect the current research priorities but also underscore the rapidly evolving landscape of multidisciplinary care for this complex patient group. However, these interventions still lack robust long-term data and results from large-scale human randomized trials, so conclusions about their efficacy and safety should be approached cautiously. Highlighting the current limitations (e.g., restricted patient populations, short follow-up, risk of complications) and clearly stating the need for future large clinical studies and extended safety data would help provide a more nuanced and critical perspective on these promising approaches.

Reply: We are proud that the Reviewer appreciated our effort to present emerging strategies, and we sincerely thank His/Her thoughtful comments and recognition of the clinical relevance of the therapeutic and prognostic strategies discussed. As suggested, the resubmitted version of our manuscript reports the need for a more critical perspective, highlighting the current limitations associated with the presented emerging approaches, specifically addressing issues such as restricted patient populations, short follow-up durations, and potential complications, as well as explicitly emphasizing the urgent need for future large-scale randomized clinical trials and extended safety data to validate these promising interventions. We believe this balanced approach provides a nuanced and realistic appraisal of the evolving therapeutic landscape, while maintaining scientific rigor and clinical relevance.
